# EdgeHOG: a method for fine-grained ancestral gene order inference at large scale

Charles Bernard[1,2,3], Yannis Nevers [1,2,4], Naga Bhushana Rao Karampudi [1,5], Kimberly J. Gilbert[1,6], Clément Train[1], Alex Warwick Vesztrocy[1,2], Natasha Glover[1,2], Adrian Altenhoff [2,7] & Christophe Dessimoz [1,2] ✉

Ancestral genomes are essential for studying the diversification of life from the last universal common ancestor to modern organisms. Methods have been proposed to infer ancestral gene order, but they lack scalability, limiting the depth to which gene neighbourhood evolution can be traced back. Here we introduce edgeHOG, a tool designed for accurate ancestral gene order inference with linear time complexity. We validated edgeHOG on various benchmarks and applied it to the entire OMA orthology database, encompassing 2,845 extant genomes across all domains of life. We reconstructed ancestral gene order for 1,133 ancestral genomes, including ancestral contigs for the last common ancestor of eukaryotes, dating back around 1.8 billion years, and observed significant functional association among neighbouring genes. EdgeHOG also dates gene adjacencies, allowing the detection of both conserved gene clusters and chromosomal rearrangements.

Modelling ancestral genomes at internal nodes of a species phylogeny is a powerful tool to trace the genetic events that shaped genome evolution. This is often done via ancestral gene repertoire reconstructions, which provide gene lists as proxies for ancestral genomes[1,2]. However, these methods do not account for gene contiguity across genomes and thus cannot capture patterns of genomic rearrangements. Ancestral gene order inference methods have emerged to fill this gap, helping detect rearrangements associated with speciation or identify functionally associated genes residing in conserved genomic neighbourhoods[3–8]. A major milestone was achieved by Muffato et al.[3] with their 'Algorithm for Gene Order Reconstruction in Ancestors' (AGORA) method and the reconstruction of gene orders for 624 ancestral genomes across five independently processed clades (200 vertebrates, 117 non-vertebrate Metazoa, 99 plants, 478 fungi and 136 protists). However, state-of-the-art methods such as AGORA rely on computationally expensive reconciled gene trees and pairwise gene order comparisons and typically struggle to process large phylogenies

with more than hundreds of genomes[9]. This scalability limitation affects both the accuracy and evolutionary scope of analyses, as including more extant genomes permits a higher resolution in reconstructing ancestral gene order and tracing their evolutionary histories further back in time. This limitation is highlighted by large-scale sequencing efforts—such as the Earth BioGenome Project[10], which aims to deliver annotated genomes for ~9,000 eukaryotic taxonomic families within the next decade—as they are rapidly outpacing the development of methods capable of harnessing the wealth of data they generate.

Here, we introduce edgeHOG, a method for reconstructing ancestral gene orders across large phylogenies while maintaining and at times even exceeding the levels of resolution and accuracy set by AGORA. Unlike approaches relying on computationally intensive reconciled gene trees, edgeHOG uses hierarchical orthologous groups (HOGs) (Fig. 1)—which are faster to infer, computable on arbitrary datasets and widely available through databases such as OMA[11], Hieranoid[12] or EggNOG[13]—to anchor comparisons of gene adjacencies between

[1]Department of Computational Biology, University of Lausanne, Lausanne, Switzerland. [2]SIB Swiss Institute of Bioinformatics, Lausanne, Switzerland. [3]Microbial Evolutionary Genomics, Institut Pasteur, Université de Paris, CNRS UMR3525, Paris, France. [4]Complex Systems and Translational Bioinformatics (CSTB), Department of Computer Science, ICube, UMR 7357, University of Strasbourg, CNRS, Strasbourg, France. [5]Department of Biology, University of Fribourg, Fribourg, Switzerland. [6]ETH Zurich, Computer Science, Zurich, Switzerland. [7]Department of Biological Sciences, SRM University, Andhra Pradesh, India. ✉e-mail: christophe.dessimoz@unil.ch

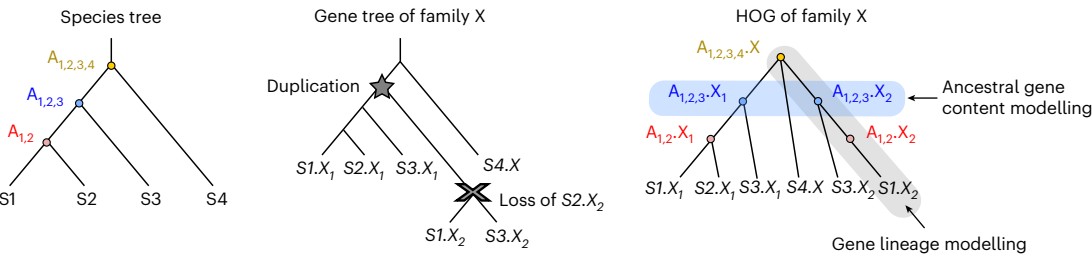

**Fig. 1 | HOGs explicitly model gene lineage, thus ancestral gene contents.**
Left: the species tree displays the evolutionary relationships between four extant species (S1, S2, S3 and S4). Its internal nodes are proxies for ancestors ($A_{1,2}$, $A_{1,2,3}$ and $A_{1,2,3,4}$). The central gene tree illustrates an example of inferred evolutionary history for a gene family (coined X) present in all species. The HOG object of the family X can be computed from the reconciled gene tree of X as in EggNOG or using fast graph-based methods as in OMA. Once pyHAM processes the HOG of family X, the graph on the right is created: each node corresponds to an extant or ancestral gene and each edge links a descendant gene to its parent gene in its most direct ancestor. When paralogues are connected to the same parental gene, it models an event of gene duplication (happening here between ancestor $A_{1,2,3,4}$ and descendant $A_{1,2,3}$). The collection of gene nodes of all HOGs' graphs at a given ancestral level eventually serves as proxies for its ancestral gene content.

genomes and propagate gene order predictions along the species tree. A proof of concept showed that HOGs are reliable for ancestral gene order inference: when using OMA-derived HOGs, AGORA reconstructed a Boreoeutherian ancestral genome similar to that inferred with Ensembl Compara's reconciled gene trees[3], available only for a restricted number of eukaryotic clades due to computational limitations[14]. Applying edgeHOG to the 2,845 extant genomes from the OMA database[11], we reconstructed ancestral gene orders for 1,133 fully browsable ancestral genomes spanning all three domains of life. This revealed a significant association between gene order and function in the last eukaryotic common ancestor (LECA), as well as intriguing patterns of chromosomal evolution, such as conserved histone gene clusters in metazoans and younger gene adjacencies on sex chromosomes across various species. EdgeHOG is available as an open-source standalone tool to process arbitrary datasets (https://github.com/DessimozLab/edgehog). In combination with the recently released FastOMA[15] software, it enables the reconstruction of ancestral gene orders for thousands of genomes (including entirely eukaryotic datasets) within a few days.

## Results

We describe edgeHOG's algorithmic principles, validation on simulated and empirical datasets and scalability and demonstrate how it enables biological insight into chromosomal evolution, including functional clustering in the LECA and patterns of gene adjacency retention across extant species and sex chromosomes.

### Algorithm overview

EdgeHOG requires a rooted species tree, the coordinates of the genes on the chromosomes or contigs (as GFF files) and the HOGs of the genomes, which can be downloaded in OrthoXML format from various orthology databases or computed from proteomes using software such as OMA Standalone[16] or FastOMA[15].

### Ancestral gene repertoire reconstruction.

HOGs can be thought of as ancestral genes, as they encompass orthologues and paralogues descending from a common ancestral gene at a specific taxonomic level (that is, internal node of the species tree)[17–19]. HOGs at a lower taxonomic level are nested within HOGs of a higher level, thereby modelling the lineage of genes, assuming strict vertical inheritance along the species tree. When distinct HOGs defined at the same taxonomic level are nested in a higher level HOG, they can be thought of as ancestral in-paralogues (Fig. 1).

### Bottom-up propagation of gene adjacencies.

Using descendant-to-parent gene links within gene lineages, observed or predicted adjacencies between two genes at a given phylogenetic level are mapped to their corresponding parental genes in the upper taxonomic level. If a gene has no parent but its flanking neighbours have one, an edge is created between these two neighbours and propagated to the upper taxonomic level, thereby modelling gene emergence and insertion events between two older genes. This process ultimately constructs a network at each level of the phylogeny, where nodes represent ancestral genes and edges link genes inferred to be of closest proximity. The weight of each edge indicates the number of propagations from descendant extant genomes (Fig. 2a).

### Top-down removal of edges not explained by parsimony.

A drawback of the bottom-up phase is that when a novel adjacency between two old genes has arisen through genomic rearrangement, propagating the adjacency to the ancestral level is essentially a mistake. Therefore, the top-down phase removes any edge propagated in ancestral synteny networks that is not supported by parsimony. This means an edge is removed if it was propagated before the last common ancestor in which the adjacency is inferred to have emerged (see Fig. 2b for details). Because the criterion of edge removal does not consider edge weights, the top-down phase is not affected by any potential tree imbalance.

### Linearization of synteny networks.

After edge removals, some ancestral genes may still have more than two neighbours due to orthology/paralogy misinferences, incorrect species tree topologies or convergent/reticulate evolution of gene adjacencies. The linearization step 'resolves' conflicting genes having more than two neighbours by selecting their two most likely flanking genes—those of maximal support (details in Fig. 2c). This results in linear ancestral contigs at each phylogenetic level, which collectively form an ancestral genome. Of note, the heuristics for determining the final linearized genome are applied independently at each internal node of the species tree, without influence from linearization choices made at other nodes.

Beyond reconstructing ancestral adjacencies, edgeHOG offers several additional features, such as predicting the orientation of genes in ancestral adjacencies, dating the age of gene adjacencies in both ancestral and extant genomes (using the information of the last common ancestor in which the adjacency is inferred to have emerged) or performing a phylostratigraphy of gene adjacency retentions, gains, losses or duplications at each internal node of the species tree.

### Extensive benchmarking on simulated and empirical data

To evaluate edgeHOG's performance in ancestral gene order reconstruction, we benchmarked it against AGORA using various simulated and real datasets (see the Supplementary Information for detailed results). In a simulation of 100 ancestral genomes, edgeHOG showed high accuracy, achieving a harmonic mean precision of 98.9% (percentage of

**Fig. 2 | EdgeHOG's algorithm. a**, The bottom-up phase. Traversing the guide species tree from leaves to root, an adjacency between two genes is propagated to a direct ancestor as long as it is inferred to have the two ancestral genes. The inferred gene gain (in grey) is accounted for by propagating an adjacency to the parental level only between the two flanking neighbours. All edges propagated to the parental level are summarized in a 'propagated graph', and the propagated form of each real edge is stored. Note that if duplicated genes or edges exist as distinct entities after the duplication event, they merge into a single gene or edge at the point of duplication and before it. **b**, The top-down phase. Traversing the tree from root to leaves, any adjacency not supported by parsimony is removed,

that is, essentially any edge supported by only one child and not by the parent (hence wrongly propagated before the last common ancestor in which the edge emerged). **c**, The linearization phase. The linearization step flags conflicting genes (those having more than two neighbours) and removes edges until the number of neighbours of a conflicting gene is no more than two. Bottom: the order in which conflicting nodes are resolved and the hierarchy of neighbours for a conflicting node are explained. The linear path weight of a neighbour is the sum of weights of all edges in the path passing by the neighbour and ending at the first node encountered with less or more than two neighbours. For each ancestor, the linearized graph constitutes edgeHOG's main output.

predicted adjacencies being correct) and a recall of 96.8% (percentage of real adjacencies being predicted). This outperformed AGORA, which showed 96.0% precision and 94.9% recall (Fig. 3a). Moreover, edgeHOG's performance was relatively consistent across all levels of the phylogenetic tree, while AGORA's accuracy slightly declined for more recent ancestors due to a bias in its weighting strategy (see the

Supplementary Information for details). In a more challenging simulation with particularly high rates of genomic rearrangement, edgeHOG outperformed AGORA more markedly, achieving a harmonic mean precision of 40.3% and recall of 18.8%, compared with AGORA's 13.9% precision and 3.8% recall (Extended Data Fig. 1 and Supplementary Information).

To benchmark on empirical data, we took advantage of the expert and thorough work performed by the Yeast Gene Order Browser (YGOB) to manually curate the likely gene order in the last common ancestor of a clade of 20 yeast species[20]. Comparing predicted gene adjacencies with that annotated by YGOB, we found that edgeHOG's precision and recall reached 91.7% and 77.5%, respectively, while AGORA's reached 90.6% and 79.2% (Fig. 3b). Notably, both tools correctly predicted gene orientations over 99% of the time.

Since YGOB may favour the method that has the most in common with its inference process, we designed an additional empirical benchmark in which we masked the gene order of ten Vertebrata species, that is, treating each gene as if it were on its own contig, and inferred them using the gene orders of 40 other Vertebrata species (see Extended Data Fig. 2 for the sampling of the 50 genomes). Specifically, we inferred the gene adjacencies of each masked genome by mapping the predicted adjacencies of its most direct ancestor onto the corresponding descendant genes, with the rationale that the number of accurate predictions projected from ancestral edges can serve as a proxy for the quality of the ancestral gene order inference. EdgeHOG again showed slightly better performance, with an average precision and recall improvement of +1.5% and +0.4%, respectively, over AGORA (Fig. 3c and see Extended Data Fig. 3 for a detailed comparison of performance across species in relation to characteristics of each masked genome (for example, contiguity level) and its corresponding ancestor (for example, phylogenetic depth)). We also found that increasing the number of genomes to 156 instead of 50 improved recall (+2.1%) (Extended Data Fig. 2) but slightly reduced precision (−0.8%) (Extended Data Fig. 4), while increasing the number of gene adjacencies in ancestral genomes (Extended Data Fig. 5). For instance, edgeHOG inferred 11,051 adjacencies for the last common ancestor of Gnathostomata with 156 species, versus 8,193 with 50 species (Extended Data Fig. 5). This demonstrates that comparing more genomes and thus handling large datasets has the potential to improve the resolution of ancestral genomes.

## Scalability

To evaluate how efficiently both tools handle large datasets, we measured their runtime (Fig. 4) and RAM usage (Extended Data Fig. 6) across eukaryotic phylogenies of increasing size. RAM usage scaled similarly for both tools, though AGORA uses ~29% less memory on average. However, differences in runtime were more pronounced. EdgeHOG's runtime scaled linearly, benefiting from its tree traversal-based edge handling, whereas AGORA's runtime inflated with larger phylogenies, due to its reliance on gene order comparisons that increase quadratically. In practical terms, edgeHOG took 1 h and 20 min to infer ancestral gene orders at each internal node of a phylogeny of 791 Eukaryotic genomes, while AGORA took 43 h and 19 min for the same task. As a result, edgeHOG currently stands as the only scalable software solution capable of reconstructing ancestral gene orders for datasets comprising thousands of eukaryotic genomes. For instance, a linear model fitted to the runtime data estimates that edgeHOG would process 10,000 eukaryotic genomes in approximately 17 h and 30 min.

## Ancestral genome orders across the three domains of life

EdgeHOG's scalability made it possible for us to process all 2,845 extant genomes in the OMA database (1,965 bacteria, 173 archaea and 707 eukaryotes), in under 3 h on a single processor. To our knowledge, this represents the largest single-run inference of ancestral gene orders using genomes across all three domains of life. The resulting collection of 1,133 ancestral genomes represents a unique resource to study ancestral synteny across key clades of the tree of life. Details for browsing this resource are outlined in the latest OMA paper[11] and a summary of the number of genes, gene adjacencies, contigs and contiguity levels for all extant and ancestral genomes of the OMA database is in Supplementary Table 1. In these resources, ancestral genomes include only genes on reconstructed contigs, excluding singleton genes (Discussion).

## Reconstruction of the LECA

The unprecedented phylogenetic depth of the analysis enabled us to reconstruct ancestral gene order in the LECA (Fig. 5; see Extended Data Fig. 7a,b for the guide species tree). EdgeHOG inferred 1,009 ancestral contigs in LECA, with lengths ranging from 2 to 19 genes (Fig. 5a). The functional similarity among genes within contigs supports the inference, consistent with the link between gene linkage and functional association[21] (Supplementary Table 2). The Gene Ontology (GO) enrichment analysis of contigs (GO terms of genes of a contig as foreground, GO terms of genes of the ancestral genome as background) confirms this trend by highlighting 194 contigs enriched in ancestral genes contributing to the same biological process (Fisher's exact test, Bonferroni-corrected $P$ value <0.05). As a sanity check, we repeated the analysis after randomizing gene order (preserving contig size), and the number of functionally enriched contigs was indeed much lower (mean of 14.6 and standard deviation of 7.9) (Fig. 5b). The tendency of neighbouring genes to be functionally related is unlikely biased by an over-representation of a gene family in multiple copies within contigs, as enriched contigs do not contain more ancestral in-paralogues than non-enriched ones (Mann–Whitney test, $P = 0.99$) (Extended Data Fig. 7c). Remarkably, reconstructed contigs contain genes that capture core pathways, with primary metabolism, translation, DNA repair and stress responses being the most represented categories of functions (Fig. 5a and Supplementary Table 2).

We computed for each LECA ancestral gene the fraction of descendants found on extant mitochondrial or chloroplast contigs (Supplementary Table 3). This showed that the long contig annotated as 'ATP synthesis' in Fig. 5a capture ancestral mitochondrion features as it contains consecutive gene adjacencies of the respiratory chain pathway (Supplementary Tables 2 and 3). However, a few contigs are erroneous based on our knowledge of eukaryotic evolution, such

**Fig. 3 | Benchmarks. a**, The simulated genome evolution benchmark. Each dot represents one of the 99 ancestral levels in a species tree with 100 extant genomes. The *x* axis gives the relative evolutionary divergence[44] from root of an ancestral node (0 for the root, near to 1 close to the leaves). The top row's *y* axis gives the precision of each algorithmic step of edgeHOG and of Agora at each ancestral level, measuring the proportion of predicted edges that are true edges in the simulated ancestral genome. The bottom row's *y* axis shows recall, that is, the proportion of true edges predicted by each method. **b**, The YGOB benchmark. The species tree depicts the phylogeny of the 20 yeast genomes in YGOB. The star indicates a whole genome duplication (WGD) event in the last common ancestor of 12 yeasts. The pink circle indicates the tree root, where YGOB-curated ancestral gene order is available. This curated order is compared with predictions by edgeHOG and AGORA, allowing evaluation of precision and recall. **c**, The masked extant gene orders benchmark. The species tree on the left shows the phylogeny of 50 vertebrate genomes from the OMA database, sampled to represent clade diversity. The ten coloured extant genomes correspond to those whose gene order is masked (and is to be inferred). The ten coloured internal levels correspond to the most direct ancestor of each masked species. Scatterplots compare edgeHOG and AGORA in inferring masked edges using a projection of each edge between two ancestral genes at the parental level onto their corresponding single-copy descendant genes in the extant masked species. Note the near 0% recall for *Salmo trutta* due to a whole gene duplication on its terminal branch, leaving no information to 'phase' the duplicated genes. Panel **c** silhouettes adapted from PhyloPic under Creative Commons licenses: *Anguilla anguilla,* Ingo Braasch (CC0 1.0); *Salmo trutta,* Carlos Cano-Barbacil (CC0 1.0); *Cirripectes variolosus*, Mykle Hoban (CC BY-SA 3.0); *Latimeria chalumnae,* Chuanixn Yu (CC0 1.0); *Podarcis muralis,* Titouan Montessuit (Attribution 4.0 International); *Crocodylus,* Becky Barnes (CC0 1.0); *Nestor notabilis,* Matías Muñoz (CC0 1.0); *Tachyglossus aculeatus,* Becky Barnes (PDM 1.0); *Manis culionensis,* Steven Traver (CC0 1.0); *Tupaia javanica,* Margot Michaud (CC0 1.0).

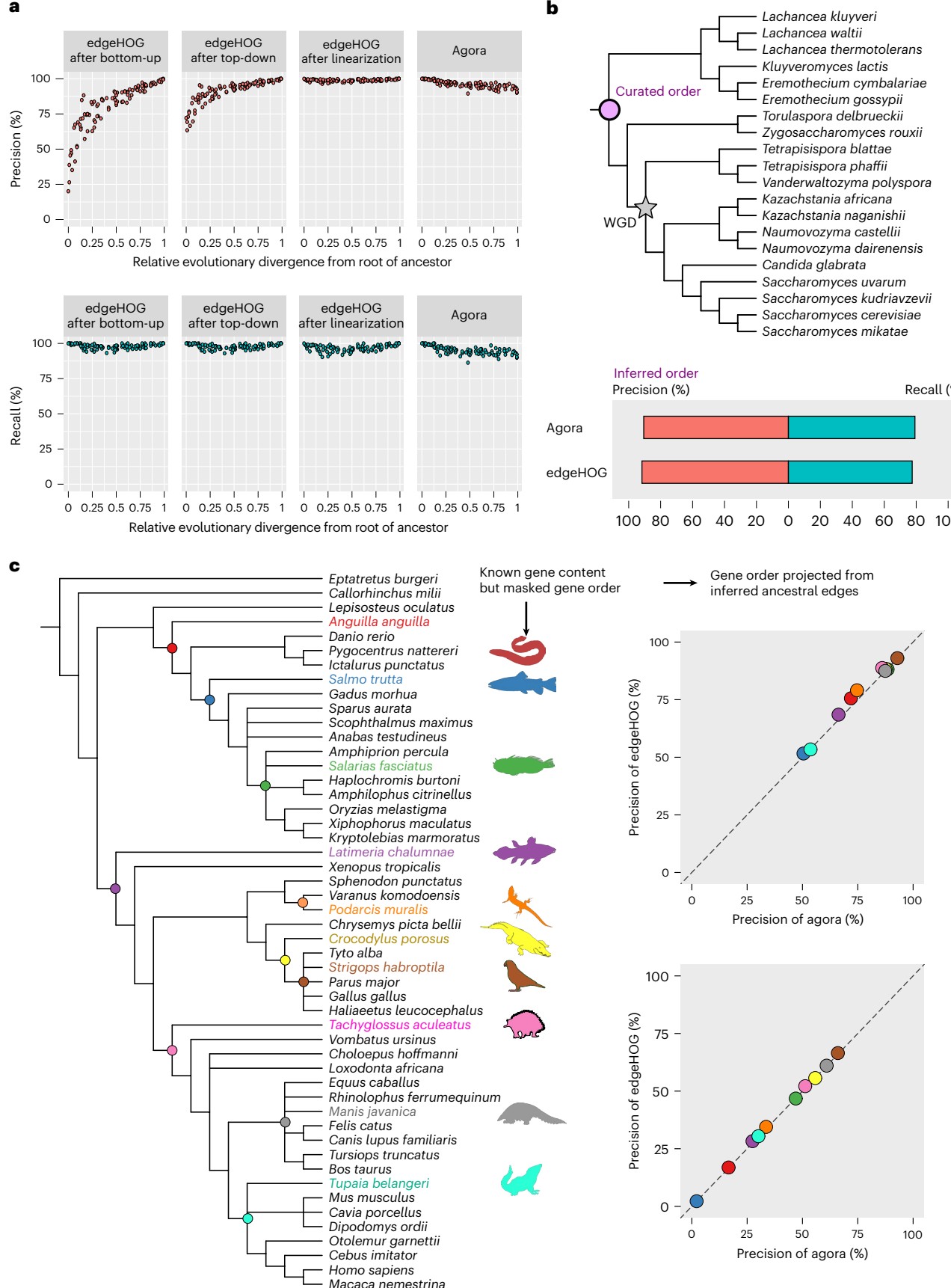

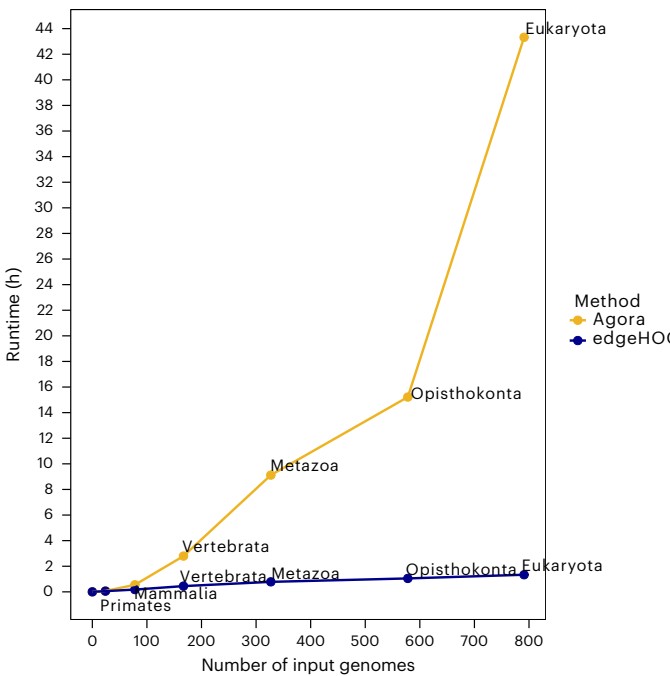

**Fig. 4 | Runtime on one processor as a function of the size of the input phylogeny.** Each dot corresponds to a clade of eukaryotic genomes from the OMA database.

as contigs containing genes involved in photosynthesis, as chloroplasts emerged from the endosymbiosis of cyanobacteria after LECA in plants, let alone the cases of secondary endosymbiosis. For instance, we found that 17 gene adjacencies in LECA were probably induced by the cyanobacterial ancestry of choloroplasts, as these edges were shared only between Cyanobacteria and Chloroplast-containing eukaryotic lineages (Supplementary Table 3). This highlights potential for future algorithmic improvements, notably accounting for reticulated evolution (Discussion).

To assess LECA's gene adjacency conservation in extant eukaryotes, we calculated the percentage of LECA edges retained per species (Supplementary Table 4, sheet 1) and the proportion of species retaining each adjacency (Supplementary Table 4, sheet 2). Extant genomes preserve 0–7.7% (1.73% average) of LECA's adjacencies (Fig. 5c), with the histone 2A–2B adjacency being the most conserved (retained in 66% of species).

### Dating gene adjacencies with edgeHOG

One novel feature of edgeHOG is the ability to assess the age of gene adjacencies of extant and ancestral genomes, that is, indicating the last common ancestor in which each adjacency is inferred to have emerged. It enables identification of conserved and divergent patterns in chromosomal organization over time. We inferred the last common ancestor (clade of origin) of all adjacencies for all eukaryotic genomes in our dataset and dated these adjacencies based on the estimated age

of their common ancestor using the TimeTree[22] resource (Fig. 6 and Supplementary Data 1). We observed remarkable patterns of chromosomal evolution in metazoan genomes.

First, a common pattern in metazoan genomes is the presence of sometimes large synteny blocks in which most adjacencies date from around 1.5 billion years (Fig. 6, blue arrows). The synteny blocks mainly comprise genes of the four subunits of the histone octamers (H2A, H2B, H3 and H4) and histone linkers (H1/H5). While paralogue adjacencies (for example, H3–H3) appear more recent, adjacencies between different histone gene families (hereafter referred to as 'histone adjacencies') are dated back to LECA (Fig. 6, *Gallus gallus*). Essentially, edgeHOG dates adjacencies between any representative of distinct gene families (HOGs) to the first occurrence of an adjacency between their common ancestors. Hence, though old adjacencies might be in multiple copies resulting from more recent tandem duplication, they are estimated as descending from ancestral single-copy adjacencies in LECA (contig with the strongest edge supports annotated as 'chromatin organization' in Fig. 5a). Histone adjacencies are common across eukaryotes, but metazoans are the only species exhibiting such clusters of histone gene adjacencies containing several copies of each subunit (Extended Data Fig. 8c) and have a higher proportion of histone adjacencies than the other clades (Extended Data Fig. 8a). This may be in part explained by metazoans having, overall, more copies of histone genes than most other eukaryotes, although this relationship is not observed in plants despite some of them having many histone copies as well (Extended Data Fig. 8b). Cluster number and size vary by species—from 12 clusters averaging 14.75 adjacencies in *Bufo bufo* to one cluster of 109 adjacencies in *Drosophila melanogaster* (Extended Data Fig. 8). Overall, our results highlight that the very old colocalization of histone subunits on the same contig in LECA (Fig. 5a) adopt a specific organization in animals where they still colocalize in the same locus but in many copies of each subunit, probably as a result of more recent tandem duplications.

Another notable pattern in adjacency ages involves sex chromosomes (teal arrows in Fig. 6). Heteromorphic sex chromosomes are pairs of homologous chromosomes that are morphologically distinct from one another, with one of them carrying a sex determination locus. They are traditionally called X and Y in species where males are heterogametic (XY) and females homogametic (XX) and Z and W when the opposite occurs (ZZ males and ZW females). These systems have been independently acquired multiple times[23]. In our dataset, heteromorphic sex chromosomes stand out as having younger adjacencies than other chromosomes (Mann–Whitney *U* test, adjacency ages on sex chromosomes versus other chromosomes in each species, alternative hypothesis: sex chromosome adjacencies are younger). As controls, we performed a similar analysis using each autosome as the focus instead. All comparison results (differences and test statistics) are given in Supplementary Table 5. We confirmed a significant trend regarding the X/Y system: both X and Y chromosomes had significantly younger adjacencies than the rest of the genome in all of the tested mammals and Diptera (chromosome X: *n* = 27, 23 mammals and 4 Diptera; chromosome Y: *n* = 12, 11 mammals and

**Fig. 5 | Functional analysis of the 1,009 ancestral contiguous regions inferred by edgeHOG in the LECA. a**, The landscape of ancestral contigs. Each dot represents an ancestral gene in a reconstructed contiguous region. The edges link the genes inferred to be mutually closest among reconstructible genes. Edge thickness is proportional to the square root of its weight (1–1,746), representing how often the edge was propagated from descendant genomes and thus the conservation level of genomic neighbourhoods. Within contigs, the black dots indicate genes whose descendant genes' GO terms contribute to enriched pathways detected in the contig. A contig with only black dots means all its genes participate in the same biological pathway. The blue dots indicate gene families with uncharacterized functions. Enriched pathways are labelled only for contigs at the figure's right border for clarity. AA, amino acids.

**b**, The number of inferred contigs with enriched pathways in LECA (first column). The graph was randomized 100 times by gene swapping, and the number of contigs with GO enrichment in each randomized graph is shown (second column). **c**, The proportion of LECA adjacencies conserved in extant eukaryotic genomes. Each dot represents an extant genome from the OMA database. The *x* axis groups genomes as animals (*n* = 275; Metazoa), fungi (*n* = 224), plants (*n* = 86; Viridiplantae or Rhodophyta) or protists (*n* = 87; other eukaryotes). The *y* axis shows the proportion of LECA's adjacencies conserved in each extant genome. The genomes are coloured by gene content completeness, assessed by OMArk. The boxplots show medians, quartiles and whiskers extending to 1.5× interquartile range beyond hinges.

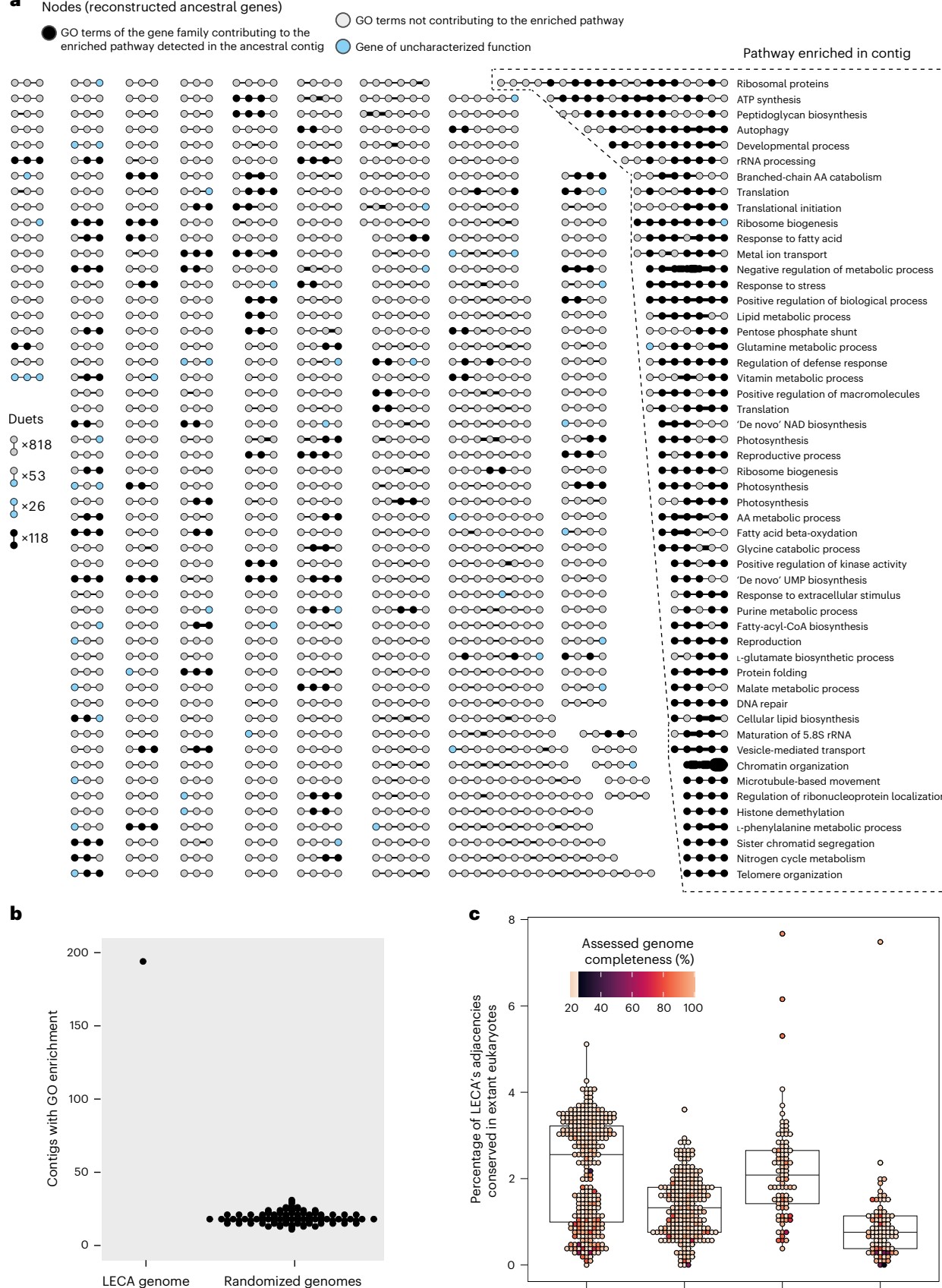

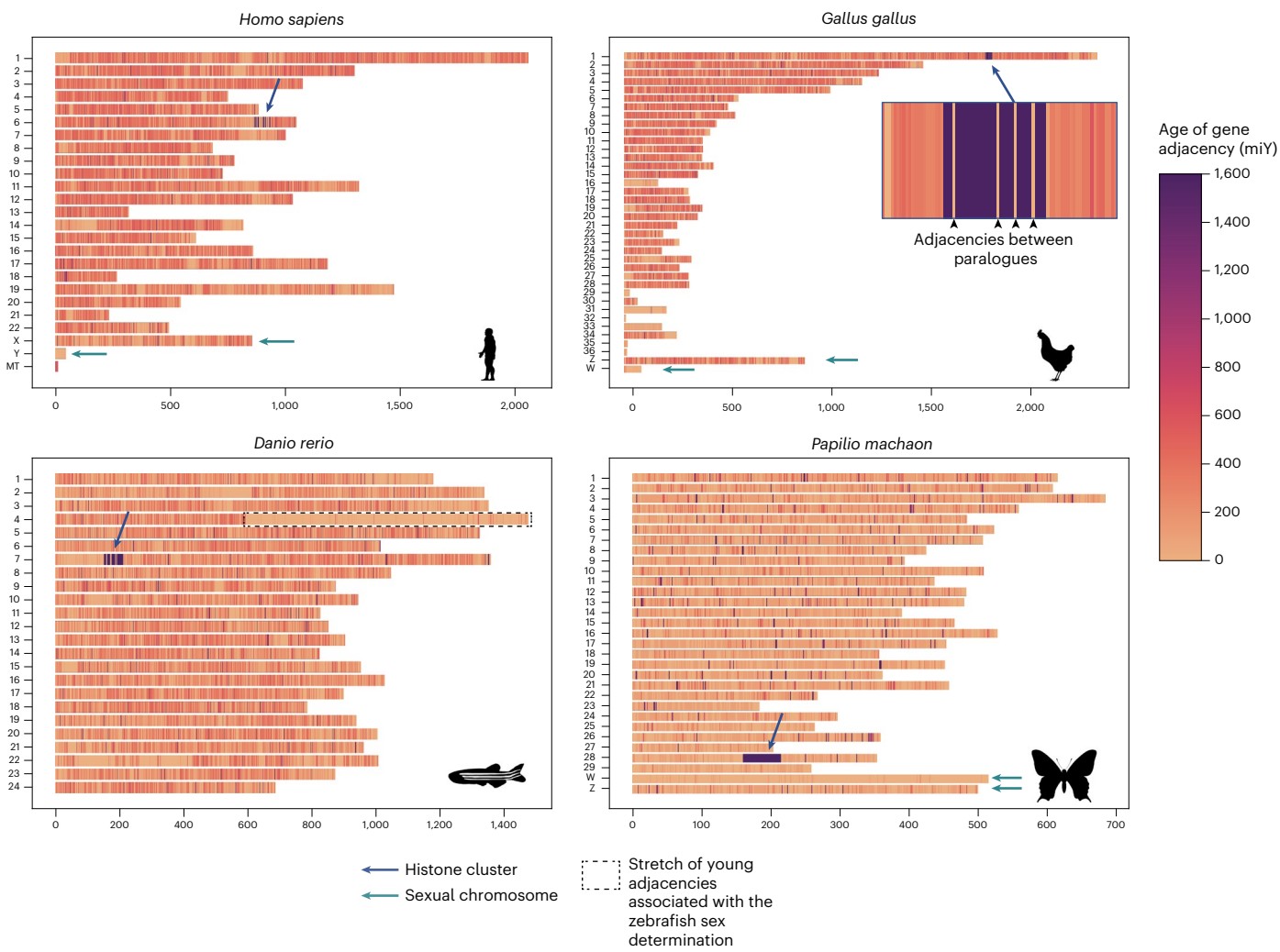

**Fig. 6 | Estimated age of gene adjacencies within chromosomes of *Homo sapiens*, *Gallus gallus*, *Danio rerio* and *Papilio machaon*.** Each subfigure corresponds to the karyotype of a given species. The *y* axis gives the name of the chromosome while the *x* axis the number of gene adjacencies. Each chromosome is represented as a stack of gene adjacencies, each coloured according to its estimated age in millions of years. The dark blue arrows indicate blocks of adjacencies dated to the eukaryotic common ancestor and enriched in adjacencies between histone genes. Squared in blue in the subfigure of *G. gallus* is a zoomed view on the histone cluster in chromosome 1: older adjacencies (dated to the eukaryotic common ancestor) are between different histone subunits while more recent adjacencies are between paralogous subunits (indicated by black triangles). The light teal arrows indicate sexual chromosomes, essentially composed of very recent adjacencies, particularly the heteromorphic chromosome (for example, Y human chromosome).

1 Diptera), except *Caenorhabditis elegans*, where no differences could be detected between the X chromosome and the autosomes. In W/Z systems, the W chromosome had significantly younger adjacencies than the rest of the genomes in all considered species ($n = 5$: 3 birds, 1 Lepidoptera and 1 Neopterygii). However, no clear pattern emerged for the Z chromosome ($n = 8$: 6 birds, 1 Lepidoptera and 1 Neopterygii), as it harboured significantly younger adjacencies only in two birds. In addition, younger adjacencies appeared on the right arm of chromosome 4 in zebrafish, where sex determination occurs, but not in other fish with homomorphic sex chromosomes. However, younger adjacencies were not unique to heterochromosomes; one or more autosomes in each species also showed significantly younger adjacencies (Supplementary Table 5), such as bird microchromosomes, primate chromosome 19 and Drosophila chromosome 4.

## Discussion

EdgeHOG unlocks key applications in comparative genomics, including tracking genomic rearrangements along a species phylogeny, identifying conserved gene clusters and improving genome assembly using gene order knowledge from other species (an example showing how ancestral gene adjacencies can expand contigs in a fragmented genome is available as a Jupyter notebook in the Figshare repository).

The ability to infer gene order conservation at large scale also facilitates the comparative genomics of fast-evolving intergenic regions, potentially identifying orthologous regulatory elements using syntenic genes to bracket non-coding regions. Likewise, it can help detect highly divergent orthologs using syntenic context, holding potential to enhance HOGs quality. Inferring HOGs, especially at deep evolutionary nodes, is challenging, often yielding more HOGs than expected ancestral genes[11]. This led us to introduce the HOG Completeness Score in the OMA browser—defined as the fraction of species in the clade represented in a HOG (ranging from 0 to 1). Low scores may indicate dubious HOGs with many inferred losses, while reliable HOGs typically score above 0.2. In the LECA reconstruction, we observed that low-score HOGs often remain as singletons, whereas high-score HOGs integrate into contigs (Extended Data Fig. 7d). Hence, edgeHOG may be useful not only to refine orthology inference but also to filter out dubious HOGs (typically excluded from reconstructed contigs).

As a powerful application illustrated in our analyses of LECA's ancestral contigs, edgeHOG can identify conserved gene clusters and highlight potential new targets for functional studies, as genes located within the same neighbourhood can be coregulated or functionally related. Moreover, tracking gene cluster evolution can offer insights into how biological functions have maintained or adapted throughout evolution.

EdgeHOG's unique option to pinpoint the clade of emergence for any gene adjacency is useful to detect evolutionary patterns of genome organization or rearrangement. For instance, it led us to recover two well-documented, outstanding patterns: histone clusters in Metazoa and relatively younger adjacencies of sex chromosomes. Histone clusters in metazoan chromosomes comprise 'blocks' of adjacencies dated by edgeHOG to LECA, consistent with the conservation of histone genes as quartet or quintet across Eukaryotes[24] and in support with the most recent suggestions of acquisitions of the histone genes as a single unit from a viral precursor[25]. Multiple clusters of histone quartets/quintets in succession is a specific feature of metazoa, probably originating from a complex history of tandem duplication and believed to be tied to the mechanisms of histone regulation in animals[24]. Dating also revealed that sex chromosomes, particularly Y, W and X, tend to have younger gene adjacencies than autosomes, reflecting known features such as higher gene turnover, gene duplication and repetitive element expansion rates in sex chromosomes than in autosomes[26], structural instability in X-specific repetitive elements[27] and rapid degeneration in Y and W due to the lack of recombination with a chromosome counterpart[28–31]. While Z chromosomes showed no clear pattern, regions such as the right arm of zebrafish chromosome 4 (rich in recent genes, pseudogenes and duplications[32]) also displayed younger adjacencies, though the link to sex determination remains unclear[33]. Younger adjacencies were also found in autosomes, including chromosome 4 in *Drosophila* (a reverted sex chromosome[34]), chromosome 19 in primates (notable for high gene density, repeats and GC content[35]) and 15 chicken microchromosomes, many with elevated repeat and GC content[36]. The ability of edgeHOG to flag the ancient origin/specific organization of histone clusters in Metazoa and chromosomal regions enriched in younger gene adjacencies highlights its potential for unravelling uncommon gene order trajectories and exploring the origin of genome architectures.

In terms of limitations, edgeHOG assumes that shared gene adjacencies across genomes are inherited from their last common ancestor, although such adjacencies can arise from horizontal gene transfer or convergent rearrangements, potentially leading to incorrect inferences, for example, photosynthetic gene contigs in LECA due to primary or secondary chloroplast endosymbiosis (Supplementary Table 3). Results in clades with reticulate evolution should thus be interpreted cautiously. Mitigation strategies include filtering for high-confidence HOGs with high completeness scores (excluding HOGs suggesting excessive gene loss) or removing edges with low weights. On another note, edgeHOG's linearized genomes prioritize microsynteny and precision and tend to have a lower contiguity level than AGORA's (Extended Data Fig. 9). While both tools can propose reconstructions of contigs of thousands to dozens of genes in ancestral species, a single missing adjacency can prevent identifying two neighbouring contigs as part of the same chromosome. For now, tools such as DESCHRAMBLER[6], which, unlike edgeHOG and AGORA, primarily optimize for contiguity may be more effective for ancestral karyotype reconstruction. Rather than bridging contigs together at the expanse of precision, future extensions of edgeHOG could group microsynteny contigs into a higher hierarchical level, for example, that of the ancestral chromosome.

Benchmarks consistently show that edgeHOG matches and even slightly exceeds AGORA in recall and precision, with better linearization near the leaves and the ability to model the emergence of a gene through dynamical reconnection of its flanking genes in the

parent graph. Most and foremost, edgeHOG scales far more efficiently than AGORA (Fig. 5). The linear scalability of edgeHOG breaks new ground and its ability to process phylogenies of thousands of genomes makes it uniquely suited to keep up with today's and tomorrow's massive sequencing projects and unlock their potential in comparative genomics.

Combining the temporal dimension of gene repertoire evolution with the spatial dimension of gene order evolution provides a comprehensive understanding of genome organization and evolutionary dynamics. Hence, our software solution opens up varied applications and advances our knowledge of genome evolution.

## Methods

### Algorithm
The detailed algorithm of edgeHOG is included in the Supplementary Information.

### Benchmarking (preparation of input data)
Input data for benchmarking are available in Supplementary Data 1. The 100 simulated lineages datasets were generated with ALF (alfsim binary version 4.0)[37], with a low mutational rate (mutRate:= 30) to facilitate downstream detection of orthologs and minimize biases inherent to orthology misinferences. Parameters regarding genome compositions and rates of gene duplications, losses and rearrangements are in Supplementary Data 1. The YGOB dataset (v7-Aug2012)[20] was downloaded from http://ygob.ucd.ie/. For the OMA Vertebrata dataset, a pruned OMA's species tree of the 50 chosen genomes was used. HOGs were derived from the tree and the all-vs-all of the 50 genomes, exported directly from the OMA browser. For all datasets, preprocessing followed the same steps. OMA Standalone inferred HOGs from the guide species tree and extant proteomes[16]. HOGs were converted to reconciled gene trees, the input format for AGORA. Gene order data (GFF files for edgeHOG, ordered gene lists for AGORA) were generated from the known extant genome structures. For ALF's output and YGOB, orders were known from metadata files. For Vertebrata, gene orders were loaded from the OMA's HDF5 file (for the 10 masked species, each gene was considered as a singleton).

### Benchmarking
EdgeHOG and AGORA (version 3.1, basic workflow[3]) were run with default parameters on all datasets. For the genome simulation benchmark, inferred adjacencies at each internal level of the species tree were compared with true adjacencies in the corresponding known ancestral genome output by ALF. For the YGOB benchmark, inferred adjacencies at the root of the species tree were compared with YGOB-curated adjacencies in this ancestor. For the masked Vertebrata species benchmarks, any adjacency between two genes in the direct ancestor of a masked extant species were propagated in the masked genome only if the two ancestral genes had each a unique descending gene in the masked genome (no descending paralogues). Projected adjacencies were then compared with real, unmasked adjacencies. For both simulated and YGOB datasets, comparing ancestral adjacencies required to perform a mapping of a modelled ancestral gene (HOG_id in edgeHOG, family_id in Agora) to the corresponding 'real' ancestral gene disclosed by ALF and YGOB. This mapping was done based on the maximal number of descending extant genes in common. For each benchmark, the recall score was computed as $100 \times TP/(TP + FN)$ and the precision as $100 \times TP/(TP + FP)$, where TP is the number of correctly inferred adjacencies, FN is the number of missed adjacencies and FP is the number of misinferred adjacencies.

### Functional analysis of LECA contigs
The LECA genome was inferred from the Nov2022 OMA release. For each Eukaryota-level HOG, ancestral GO terms were assigned as the union of its extant descendants' terms. A Gene Ontology Enrichment

Analysis was performed using goatools version 1.3.1 (ref. [38]), with contig HOGs as the foreground and all Eukaryota-level HOGs as background. Enriched terms were those with Bonferroni-adjusted $P < 0.05$ (Fisher's exact test). Randomized graphs were generated by swapping HOGs among the collection of contigs, which affected only the gene content of contigs and not their topology. Contigs were visualized with Cytoscape version 3.10.0 (ref. [39]).

### LECA's adjacencies conservation in eukaryotes

Using pyHAM 1.2.0 (ref. [40]), we retrieved all descendant genes per species for each HOG on LECA's contigs. For each ancestral adjacency, we checked extant synteny graphs in species where both ancestral genes had descendants. If an adjacency existed between any descendant genes, the extant adjacency was considered conserved in that species.

### Dating gene adjacencies

The taxon of origin for gene adjacencies was determined using Edge-HOG's date_edges option. Taxon ages were obtained from TimeTree[22] (https://timetree.org) by uploading species lists from the OMA Database. Since TimeTree lacks some OMA species, we first considered a reduced OMA Taxonomy containing only species shared with TimeTree and attributed an age of all non-conflicting internal nodes between OMA and TimeTree. Finally, we attributed an age of 0 to any leaf in the OMA Taxonomy. Any node left with no age at this point was assigned the average age of its most recent ancestor and its oldest child with age info. A companion script for dating gene adjacencies with TimeTree is available in the EdgeHOG GitHub repository. Histone adjacency clusters were defined as groups with over four adjacencies between histone genes from distinct HOGs and fewer than ten genes separating these adjacencies. The cluster size equals the number of histone adjacencies within the cluster. For the sex chromosomes, we selected genomes with clearly identified sex chromosomes (X, Y, Z and W) or numbered chromosomes from the OMA Database, excluding fungi with Roman numeral chromosomes. Only canonical chromosomes (numbers or letters) were considered, excluding incomplete contigs and scaffolds. Comparisons were made between each sex chromosome and all other complete chromosomes, as well as each autosome against the other chromosomes. One-sided Mann–Whitney tests assessed whether the distribution of adjacency ages was similar between the sex and the other chromosomes, with the alternative hypothesis being the sex chromosome having younger adjacencies than the others. The $P$ values were adjusted for multiple testing based on the number of chromosomes per species.

### Reporting summary

Further information on research design is available in the Nature Portfolio Reporting Summary linked to this article.

## Data availability

Supplementary Data 1 (data available via Figshare at https://doi.org/10.6084/m9.figshare.26425081.v2, ref. [41]) contains all the scripts and datasets (simulations, YGOB and OMA Vertebrata species) used for the benchmarking of edgeHOG. It all also contains the data and scripts used for downstream analyses, that is, the functional annotation of reconstructed contigs in LECA, the sequences of the HOGs in LECA's contigs having more than two genes, the study of the conservation of LECA's adjacencies in extant eukaryotes and the dating of gene adjacencies in extant eukaryotes (along with plots with dated adjacencies for 706 eukaryotic genomes)[42]. Reconstructed ancestral gene orders are browsable via OMA browser at https://omabrowser.org/oma/genome/.

## Code availability

EdgeHOG is free open-source software (MIT license) available via GitHub at https://github.com/DessimozLab/edgeHOG (also available via Figshare at https://doi.org/10.6084/m9.figshare.29378213, ref. [43]).

The code and scripts used in the analyses of this Article are available in Supplementary Data 1 (ref. [42]).

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

## Acknowledgements

The project was supported by Swiss National Science Foundation (grant nos. 183723 and 205085 to C.D.).

## Author contributions

C.B. designed the top-down and linearization phases of edgeHOG, implemented the software, performed the benchmarking, conducted the downstream analyses and wrote the manuscript with input from all coauthors. N.B.R.K. contributed to the design and the code of the ALF and YGOB benchmarks. K.J.G. designed the bottom-up phase of edgeHOG. A.W.V. contributed to the ancestral GO enrichment analysis. C.T. implemented pyHAM and wrote the code to explore and visualize ancestral gene orders on the OMA browser. Y.N. assessed the quality of ancestral adjacency reconstructions and designed, performed and interpreted the adjacency dating analyses. N.G. participated in the design of the full study, contributed to the manuscript and contributed to project supervision. A.A. contributed to the code of the preprocessing, bottom-up and outputting steps of edgeHOG and contributed to the design and implementation of the benchmarking protocoles and in the integration of edgeHOG in the ecosystem of tools of the OMA browser. C.D. conceptualized and supervised the project.

## Competing interests

The authors declare no competing interests.

## Additional information

**Extended data** is available for this paper at https://doi.org/10.1038/s41559-025-02818-0.

**Correspondence and requests for materials** should be addressed to Christophe Dessimoz.

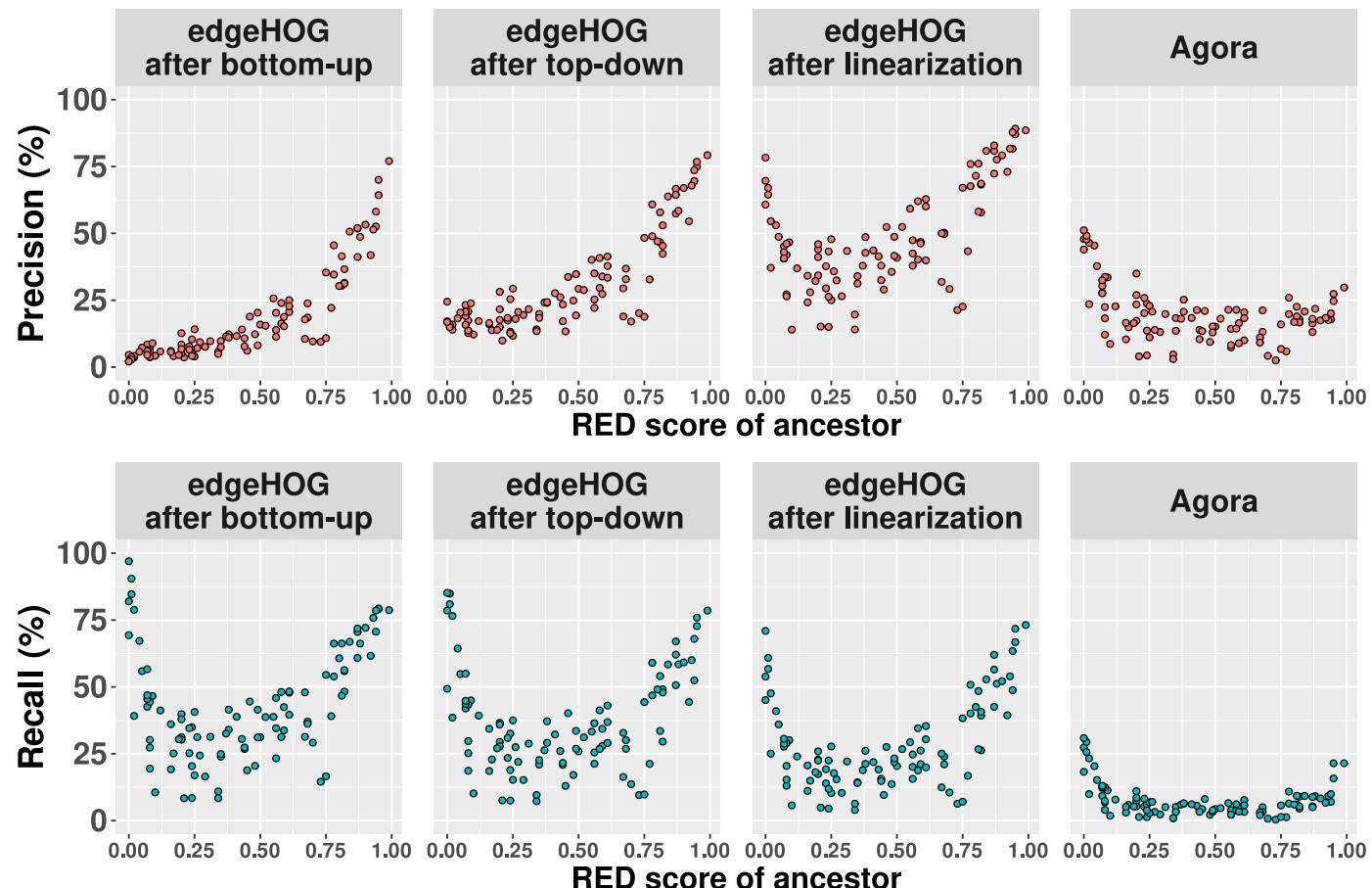

**Extended Data Fig. 1 | 'Difficult' Simulation of genome evolution.** Each dot represents one of the 99 ancestral levels in a species tree with 100 extant genomes. The x-axis gives the *Relative Evolutionary Divergence* of an ancestral node (0 for the root, near to 1 close to the leaves). The top row's y-axis gives the precision of each algorithmic step of edgeHOG and of Agora at each ancestral level, measuring the proportion of predicted edges that are true edges in the simulated ancestral genome. The bottom row's y-axis shows recall, *that is*, the proportion of true edges predicted by each method. Parameters of the simulation are given in the Supplementary Information.

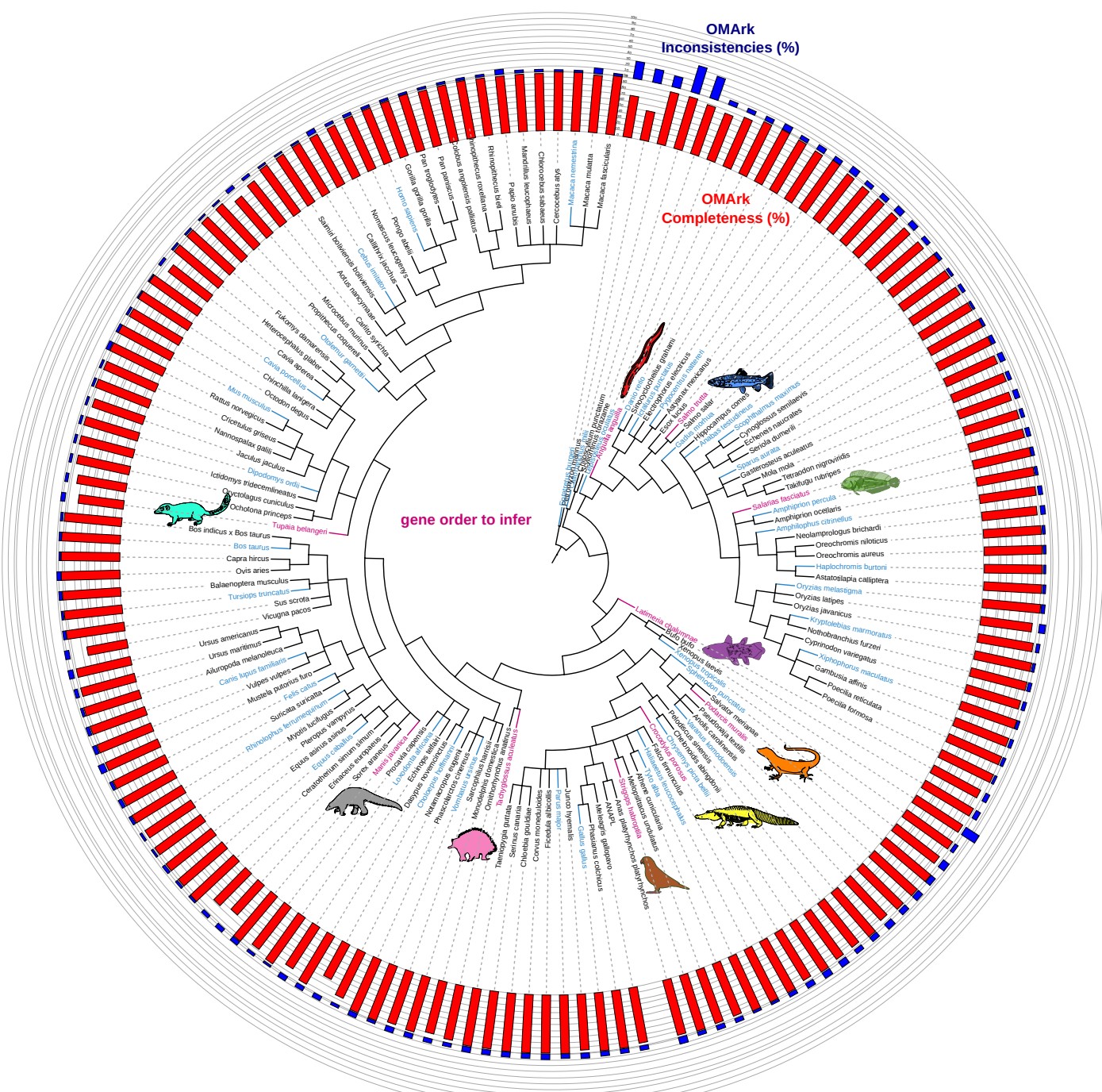

**Extended Data Fig. 2 | Selection of the 10 masked genomes (pink) and the 40 other representative genomes (blue) within the Vertebrata clade in OMA.** The bars give the OMArk-assessed completeness and the consistency of gene repertoire of each genome relative to the closest species in OMA.

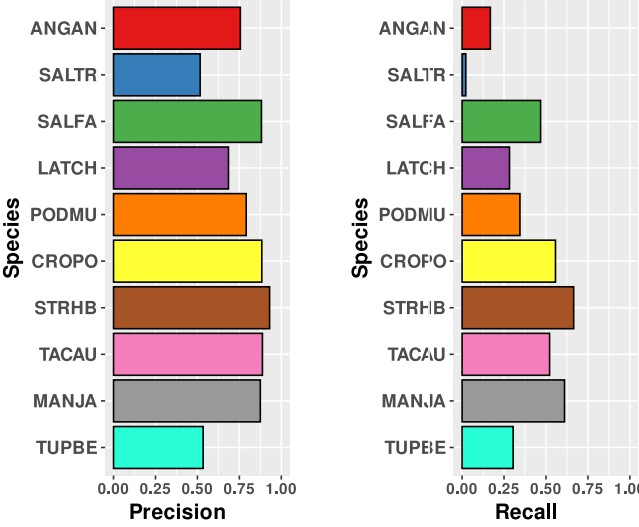

## Extant masked genome's characteristics

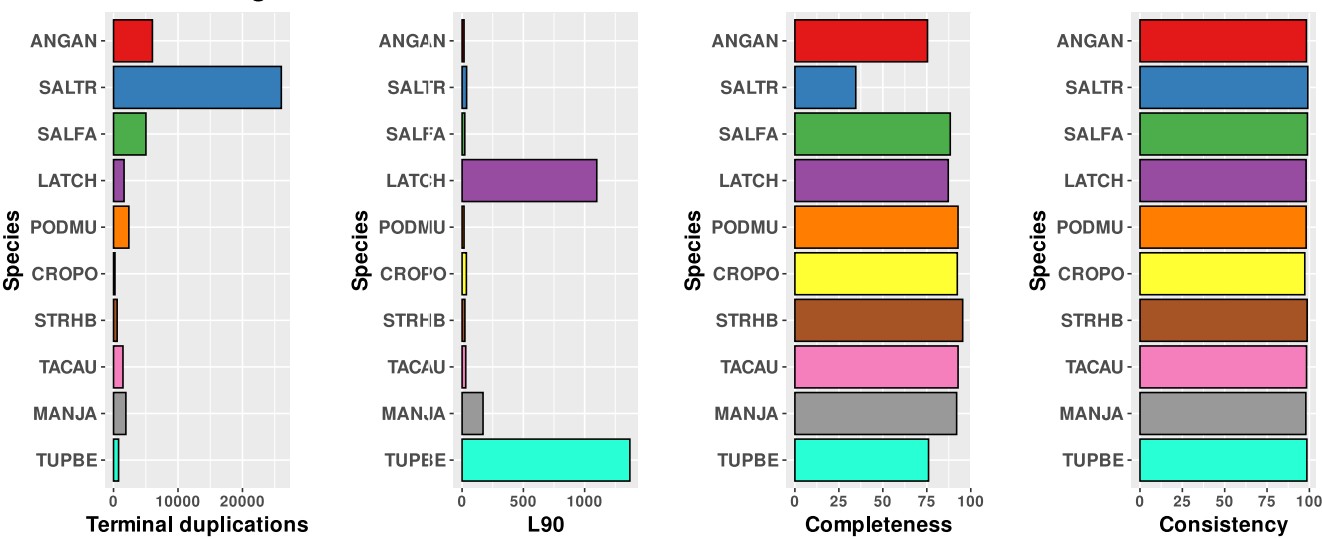

## Direct ancestor's characteristics

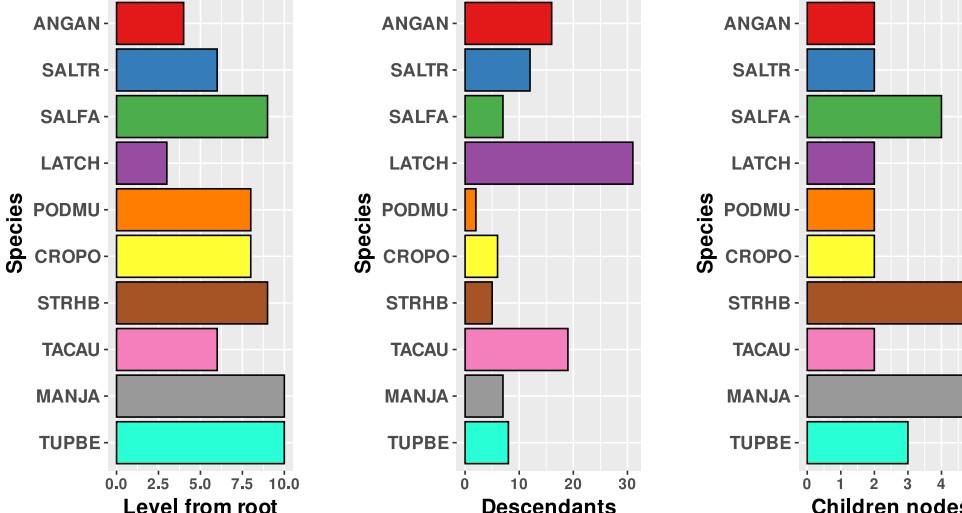

**Extended Data Fig. 3 | See next page for caption.**

**Extended Data Fig. 3 | Recall and Precision for reconstructing a masked extant gene orders based on the inferred gene order of its direct ancestor (first row), alongside characteristics of the masked species (second row) and of its most direct ancestor (third row).** Terminal duplications represent the number of gene duplications inferred by OMA after the direct ancestor. L90 indicates the minimal number of contigs required to capture 90% of the genes in the masked genome (*that is*, lower L90 values indicate higher assembly contiguity). Genome completeness, assessed by OMArk, estimates the proportion of expected genes compared to related species in OMA. Genome consistency measures the proportion of true positive genes in the proteome, using comparisons to related species as a proxy. The level from root of the direct ancestor refers to the number of parental nodes required to reach the ancestor from the root. Number of descendants refers to the number of leaves descending from the ancestral node, while the number of child nodes indicates the polytomy level of the node.

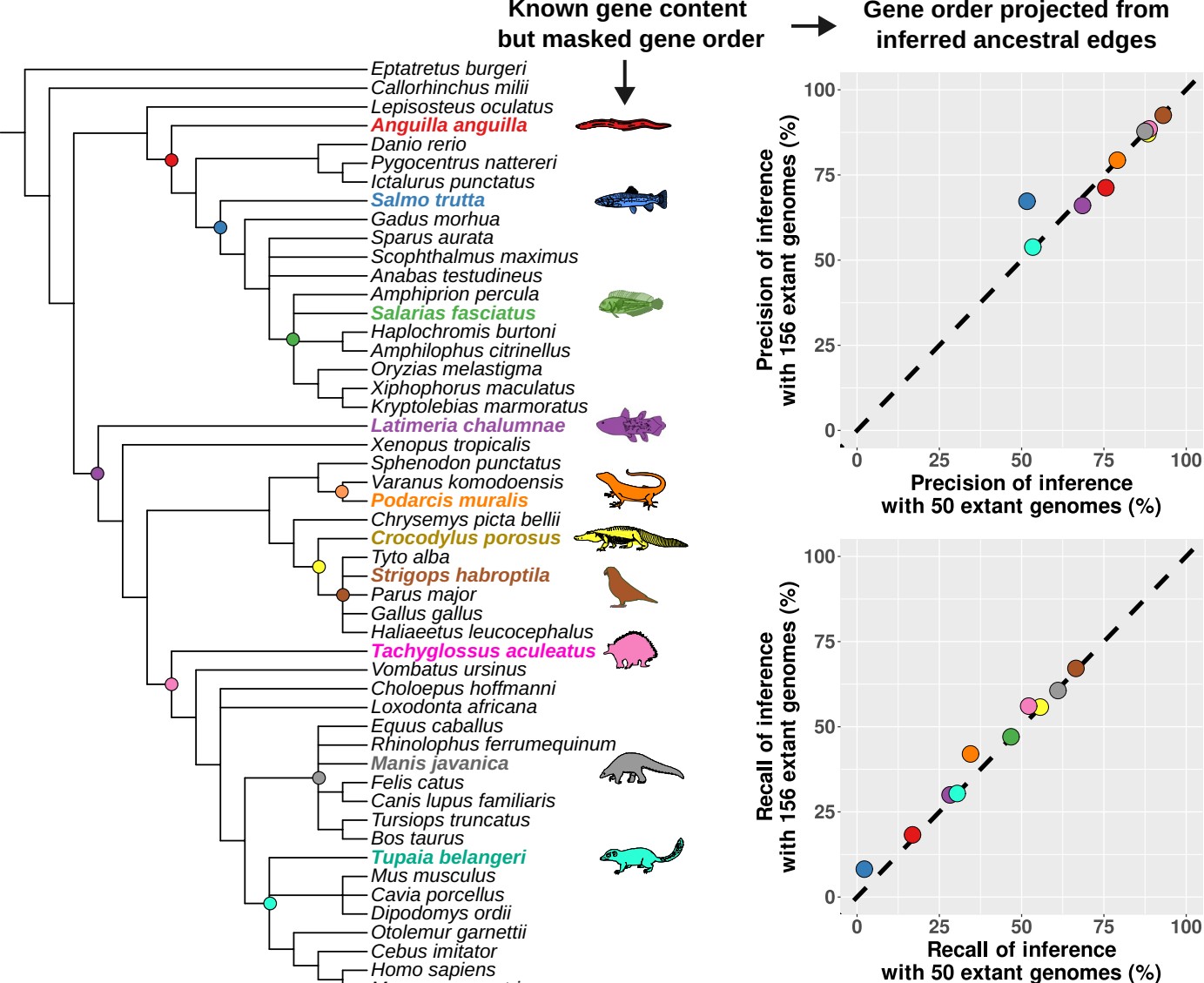

**Extended Data Fig. 4 | Recall and Precision of edgeHOG for reconstructing masked gene orders when using 50 extant species (X axes) or 156 extant species (Y axes).** The species tree on the left corresponds to the phylogeny of 50 extant genomes. The 10 colored extant genomes correspond to those whose gene order is masked (and is to be inferred). The 10 colored internal levels correspond to the most direct ancestor of each masked species from which the predicted gene order in the masked species is propagated from. The phylogeny of the 156 genomes is displayed in Extended Data Fig. 2.

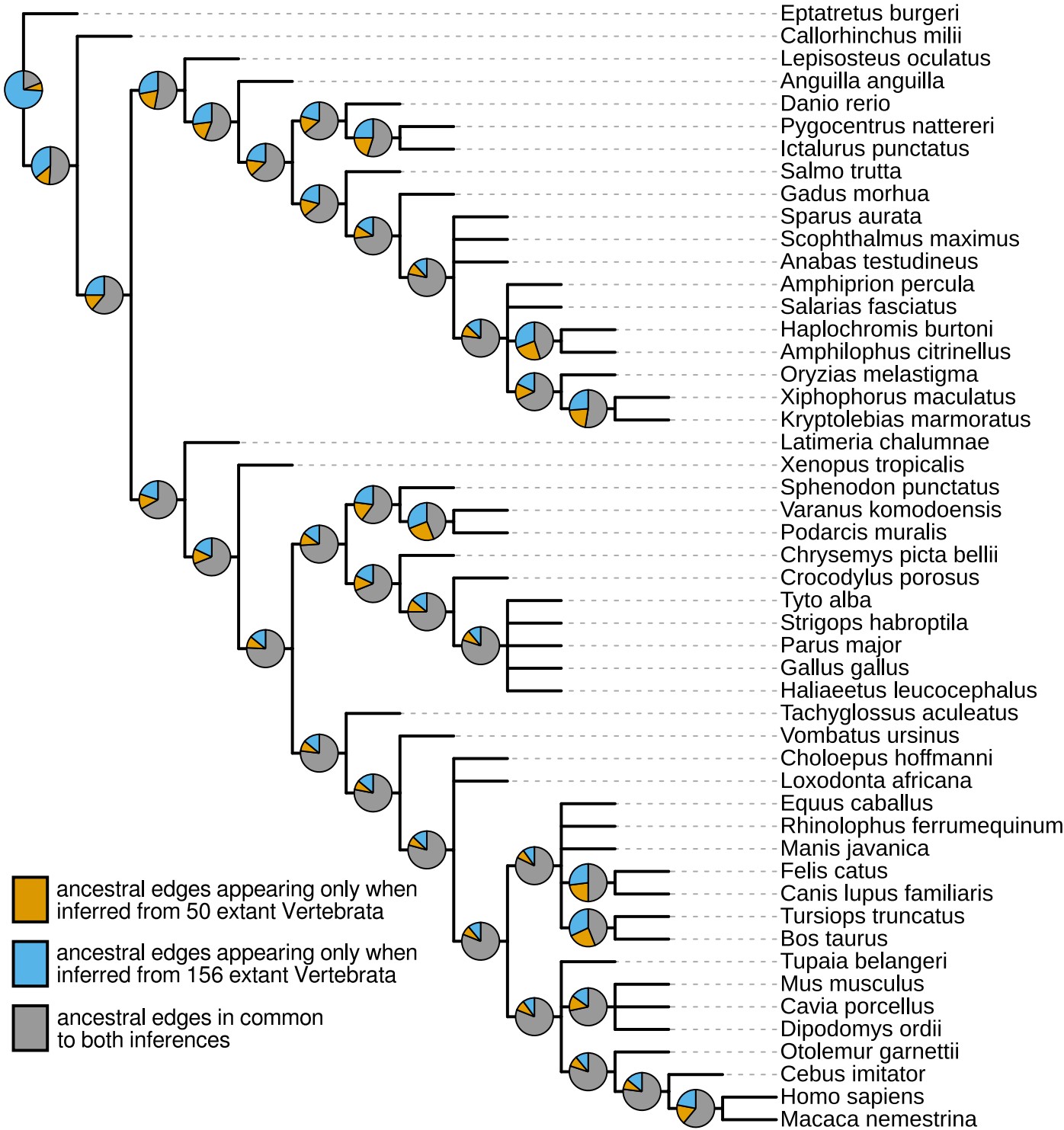

**Extended Data Fig. 5 | Similarity of ancestral reconstructions when using either 50 extant Vertebrata species or 156 species.** Both phylogenies of 50 and 156 species have 36 ancestral nodes in common. For each ancestor in common, the pie chart gives the proportion of gene adjacencies in common to both inferences (grey), specific to the 50 species dataset (yellow) or to the 156 species dataset (blue).

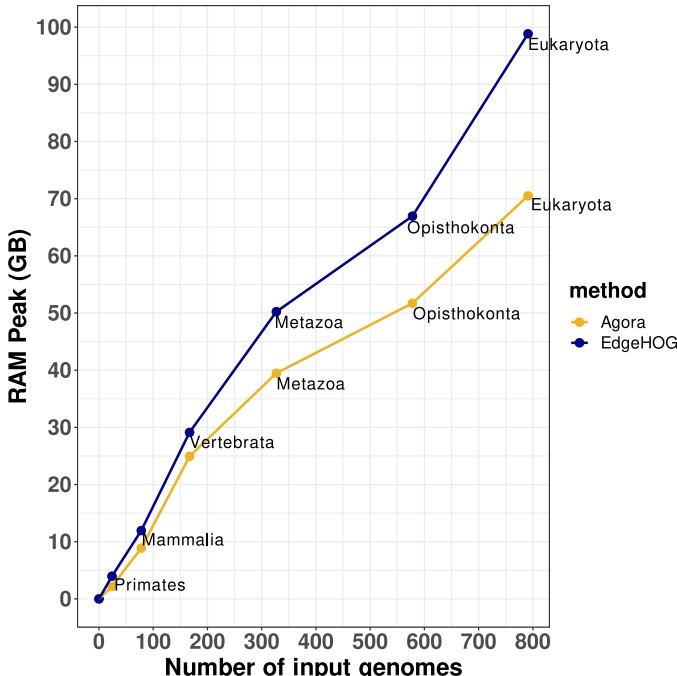

**Extended Data Fig. 6 | Peak RAM usage (in GB) of edgeHOG and AGORA as a function of the size of the input phylogeny.** Each dot corresponds to a clade of eukaryotic genomes from the OMA database.

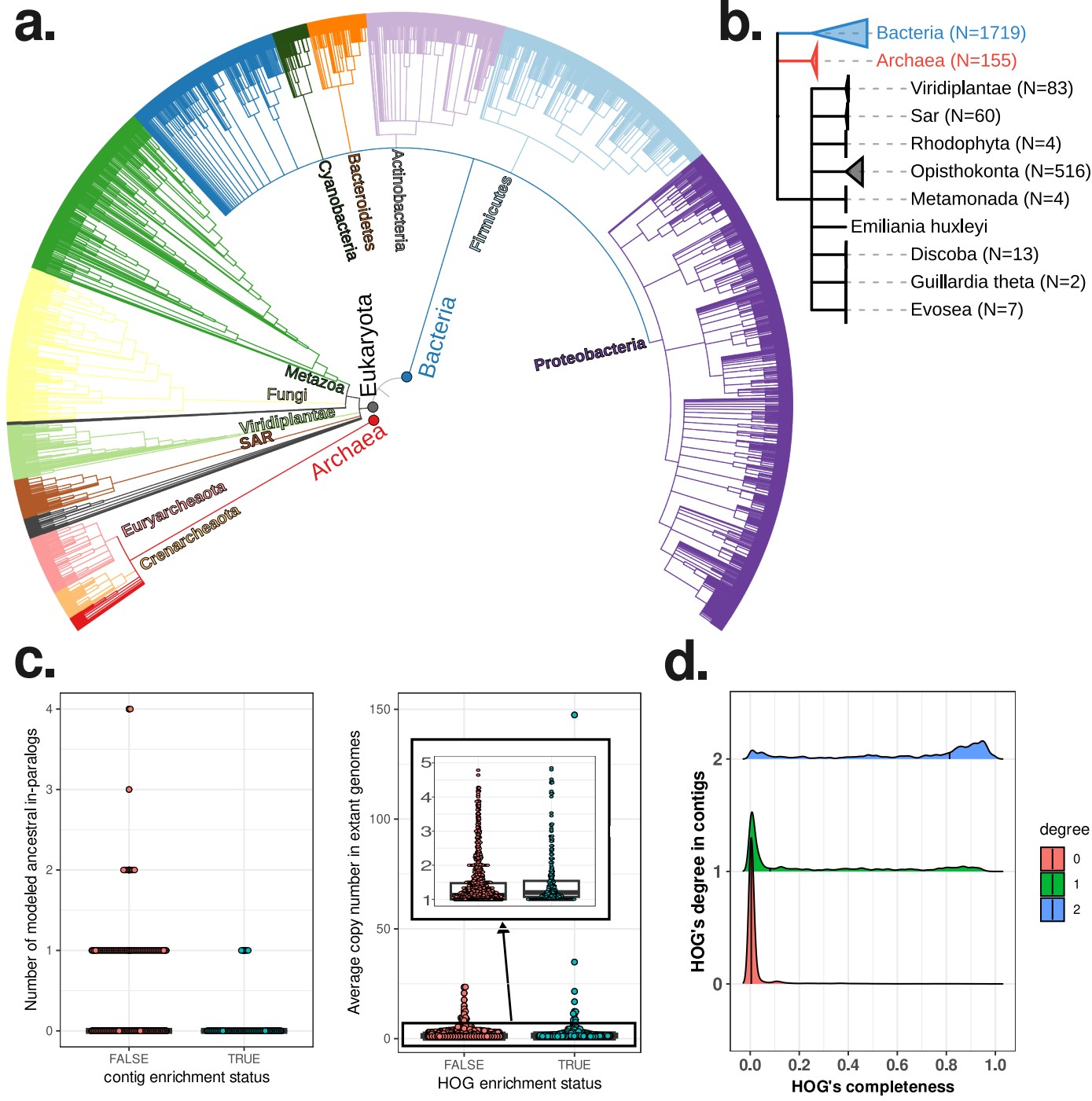

**Extended Data Fig. 7 | Information on LECA's reconstructed gene order.**
**a. Guide species tree for LECA reconstruction**. This corresponds to the species
tree of the OMA database version Nov2022, essentially a pruned version of
the NCBI taxonomy tree, containing only the genomes present in OMA. **b. The
LECA node is a polytomy with 9 children nodes and has 2 outgroups** (Archaea
and Bacteria). **c. Impact of gene duplication on biological process GO term
enrichment of LECA's contigs**. The distribution on the left shows the number
of modeled ancestral in-paralogs in contigs with and without biological process
GO term enrichment. Contigs without enrichment are indicated in red (n = 816),
while those with enrichment are shown in blue (n = 193). There is no significant
difference in the number of modeled ancestral in-paralogs between enriched and
non-enriched contigs (Mann-Whitney test, alternative = 'greater', p-value = 0.99).
The distribution on the right displays the average number of descendant in-
paralogs in extant species for each HOG/ancestral gene within a contig (with a

zoom in in the region with the highest density of points), grouped by whether the
ancestral gene's GO terms contribute to the enrichment of the contig's GO terms
(n = 349) or not (n = 1548). HOGs associated with GO term enrichment tend to
exhibit a slightly higher number of descendant in-paralogs (Mann-Whitney test,
alternative = 'greater', p-value = 2.2e-16; median = 1.22 for enriched HOGs, median
= 1.13 for others). **d. Relationship between the degree (number of neighbors)
of HOGs in LECA's contigs and their Completeness Score**. The plot gives the
distribution of Completeness Scores for HOGs at the LECA level in the current
OMA release that are included within reconstructed contigs (degree=2; n = 1139),
that are terminal genes in contigs (degree=1; n = 1848), that are singletons and
thus excluded from the ancestral genome (degree=0; n = 37773). The vertical
line in each ridge plot gives the median of the distribution for each degree
level. It shows that the most reliable HOGs are included within contigs and that
singletons typically correspond to low quality HOGs.

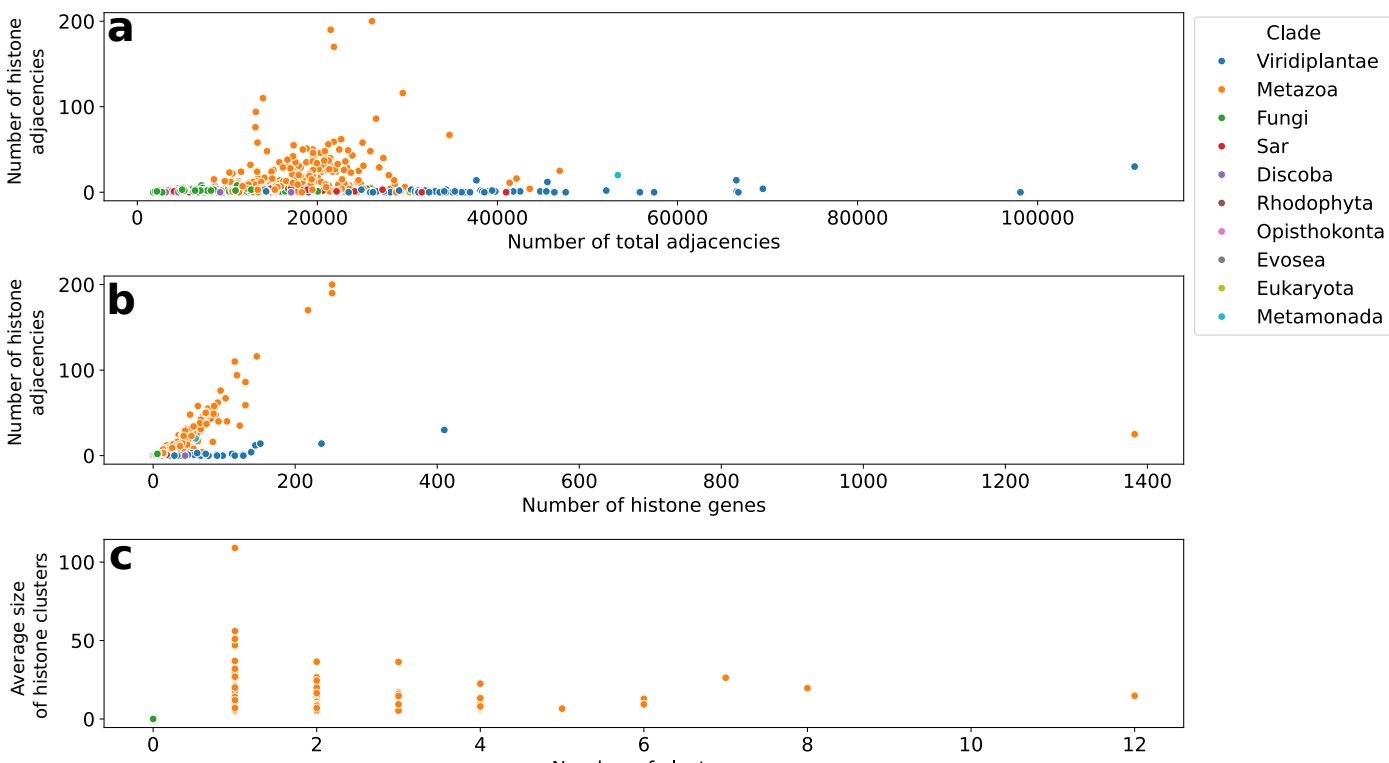

**Extended Data Fig. 8 | Histone clusters in eukaryotic clades. a. Adjacencies between histone genes are more prevalent in Metazoa**. Number of adjacencies between distinct histone genes as a function of number of total genewise adjacencies. **b. Number of histone adjacencies is proportional to the number of histone genes in Metazoa**. Number of adjacencies between distinct histone genes as a function of the number of histone genes. **c. Organization of histones in gene clusters in Metazoa**. Number of distinct histone clusters (x) and average number of adjacencies between histone genes within it (y). Colors indicate the eukaryotic clade to which each species belongs to.

## 112 ancestral genomes (Vertebrata Clade)

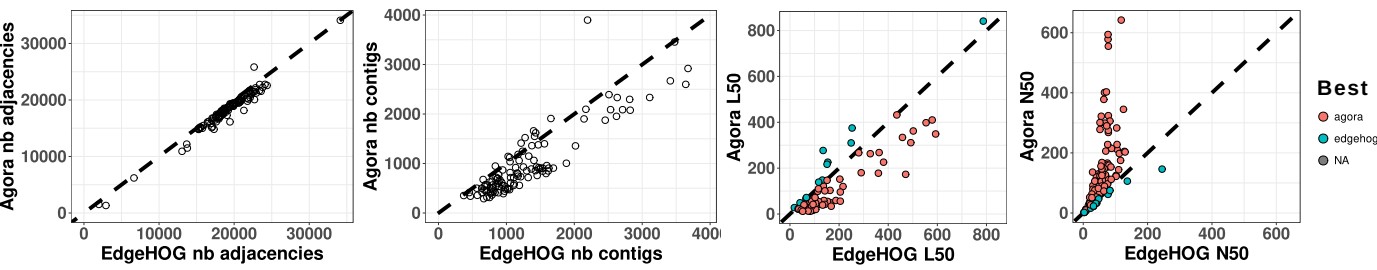

## 19 ancestral genomes (Yeast Gene Order Browser Clade)

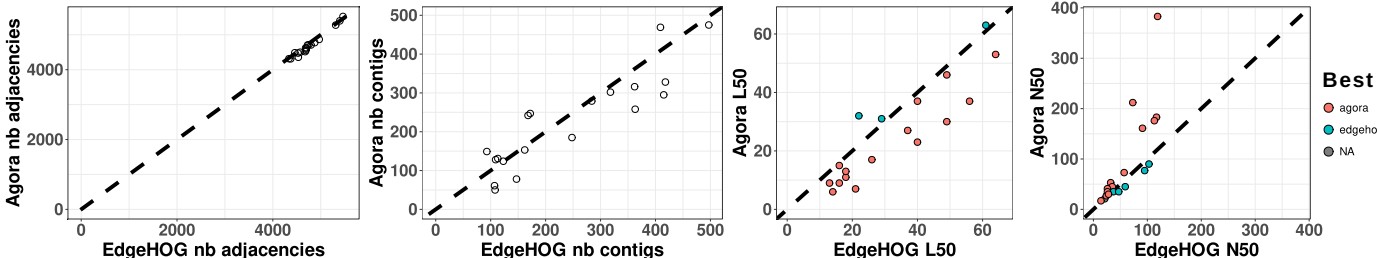

**Extended Data Fig. 9 | Comparison of contiguity levels between AGORA and edgeHOG ancestral genomes.** Each point on the scatterplots represents an ancestral genome. The first row shows inferences for the Vertebrata clade from OMA, while the second row represents yeast clades from YGOB. The first column compares the number of gene adjacencies predicted by both tools, the second compares the number of contigs, the third compares the L50 (that is, the minimum number of longest contigs needed to cover half of the predicted adjacencies for an ancestor, based on the smaller of the total adjacencies predicted by AGORA and EdgeHOG), and the last column shows the N50, which is the length of the Nth longest contig required to cover half of the adjacencies.

# Reporting Summary

## Statistics

For all statistical analyses, confirm that the following items are present in the figure legend, table legend, main text, or Methods section.

| n/a | Confirmed | |
|---|---|---|
| ☐ | ☒ | The exact sample size (*n*) for each experimental group/condition, given as a discrete number and unit of measurement |
| ☐ | ☒ | A statement on whether measurements were taken from distinct samples or whether the same sample was measured repeatedly |
| ☐ | ☒ | The statistical test(s) used AND whether they are one- or two-sided *Only common tests should be described solely by name; describe more complex techniques in the Methods section.* |
| ☒ | ☐ | A description of all covariates tested |
| ☐ | ☒ | A description of any assumptions or corrections, such as tests of normality and adjustment for multiple comparisons |
| ☐ | ☒ | A full description of the statistical parameters including central tendency (e.g. means) or other basic estimates (e.g. regression coefficient) AND variation (e.g. standard deviation) or associated estimates of uncertainty (e.g. confidence intervals) |
| ☐ | ☒ | For null hypothesis testing, the test statistic (e.g. *F*, *t*, *r*) with confidence intervals, effect sizes, degrees of freedom and *P* value noted *Give P values as exact values whenever suitable.* |
| ☒ | ☐ | For Bayesian analysis, information on the choice of priors and Markov chain Monte Carlo settings |
| ☒ | ☐ | For hierarchical and complex designs, identification of the appropriate level for tests and full reporting of outcomes |
| ☒ | ☐ | Estimates of effect sizes (e.g. Cohen's *d*, Pearson's *r*), indicating how they were calculated |

*Our web collection on statistics for biologists contains articles on many of the points above.*

## Software and code

Policy information about availability of computer code

| Data collection | n/a |
|---|---|
| Data analysis | In benchmarks, HOGs inference was performed with OMA standalone version 2.6.0 (https://github.com/DessimozLab/OmaStandalone). Ancestral gene order inferences were performed with edgeHOG version 0.1.0 (https://github.com/DessimozLab/edgehog) and AGORA basic workflow version 3.1 (https://github.com/DyogenIBENS/Agora). Functional characterisation of ancestral genes/HOGs (e.g. fraction of descendant genes on organelle contigs, average copy number in extant genomes etc.) was performed using pyHAM version 1.2.0 (https://github.com/DessimozLab/pyham). Gene Ontology Enrichment Analysis (GOEA) in LECA's contigs was performed using goatools version 1.3.1 (https://github.com/tanghaibao/goatools). LECA's contigs were visualized with Cytoscape version 3.10.0 (https://cytoscape.org/). Dating of adjacencies in MiY was performed using the protocole described in https://github.com/DessimozLab/edgehog/blob/main/README.md. All scripts used in the study are available in the Supplementary Dataset 1: https://doi.org/10.6084/m9.figshare.26425081.v2 |

For manuscripts utilizing custom algorithms or software that are central to the research but not yet described in published literature, software must be made available to editors and reviewers. We strongly encourage code deposition in a community repository (e.g. GitHub). See the Nature Portfolio guidelines for submitting code & software for further information.

## Data

Policy information about <u>availability of data</u>

All manuscripts must include a <u>data availability statement</u>. This statement should provide the following information, where applicable:

- Accession codes, unique identifiers, or web links for publicly available datasets
- A description of any restrictions on data availability
- For clinical datasets or third party data, please ensure that the statement adheres to our <u>policy</u>

> Simulated ancestral and extant genomes have been generated with ALF (alfsim binary version 4.0, http://alfsim.org), using parameters listed in the Supplementary Dataset 1: https://doi.org/10.6084/m9.figshare.26425081.v2.
> The Yeast Gene Order Browser dataset v7-Aug2012 was downloaded from http://ygob.ucd.ie/
> All other genomes used in this study are from the OMA database : https://omabrowser.org/oma/home/ (using the OMA Browser database HDF5 file, the species tree in newick and the HOGs orthoxml file available in https://omabrowser.org/oma/archives/All.Nov2022/ and https://omabrowser.org/oma/archives/All.Jul2023/)
> Ages in MiY of ancestors were fetched from TimeTree version 5 resource : https://timetree.org
> The predicted ancestral gene orders are browsable in https://omabrowser.org/oma/genome/
> Data used in analyses are provided in the Supplementary Dataset 1: https://doi.org/10.6084/m9.figshare.26425081.v2

## Research involving human participants, their data, or biological material

Policy information about studies with <u>human participants or human data</u>. See also policy information about <u>sex, gender (identity/presentation), and sexual orientation</u> and <u>race, ethnicity and racism</u>.

| | |
|---|---|
| Reporting on sex and gender | n/a |
| Reporting on race, ethnicity, or other socially relevant groupings | n/a |
| Population characteristics | n/a |
| Recruitment | n/a |
| Ethics oversight | n/a |

Note that full information on the approval of the study protocol must also be provided in the manuscript.

# Field-specific reporting

Please select the one below that is the best fit for your research. If you are not sure, read the appropriate sections before making your selection.

☐ Life sciences    ☐ Behavioural & social sciences    ☒ Ecological, evolutionary & environmental sciences

For a reference copy of the document with all sections, see nature.com/documents/nr-reporting-summary-flat.pdf

# Ecological, evolutionary & environmental sciences study design

All studies must disclose on these points even when the disclosure is negative.

| | |
|---|---|
| Study description | This study presents edgeHOG, a computational tool for fine-grained ancestral order inference at large scale. Utilizing publicly available genomic data, we conducted large-scale analyses which were validated through simulated and empirical data as described in the Methods section. The notion of traditional experimental units or replicates is not applicable to this work |
| Research sample | 2845 genomes from the Jul2023 release of the OMA database (1965 bacteria, 173 archaea, 707 eukaryotes). Validation was performed on smaller datasets as described in the methods section. |
| Sampling strategy | All available genomes in the OMA database were used to reconstruct ancestral gene orders. |
| Data collection | n/a |
| Timing and spatial scale | n/a |
| Data exclusions | No data was excluded |
| Reproducibility | Input data, code, and scripts are provided. |
| Randomization | n/a |

| Blinding | n/a |
|---|---|

Did the study involve field work? ☐ Yes ☒ No

# Reporting for specific materials, systems and methods

We require information from authors about some types of materials, experimental systems and methods used in many studies. Here, indicate whether each material, system or method listed is relevant to your study. If you are not sure if a list item applies to your research, read the appropriate section before selecting a response.

## Materials & experimental systems

| n/a | Involved in the study |
|---|---|
| ☒ ☐ | Antibodies |
| ☒ ☐ | Eukaryotic cell lines |
| ☒ ☐ | Palaeontology and archaeology |
| ☒ ☐ | Animals and other organisms |
| ☒ ☐ | Clinical data |
| ☒ ☐ | Dual use research of concern |
| ☒ ☐ | Plants |

## Methods

| n/a | Involved in the study |
|---|---|
| ☒ ☐ | ChIP-seq |
| ☒ ☐ | Flow cytometry |
| ☒ ☐ | MRI-based neuroimaging |

## Plants

| Seed stocks | n/a |
|---|---|

| Novel plant genotypes | n/a |
|---|---|

| Authentication | n/a |
|---|---|

