## [Peer Review File · Nature Ecology & Evolution]

EdgeHOG: a method for fine-grained ancestral gene order inference at large scale

Corresponding Author: Professor Christophe Dessimoz

Version 0:

Decision Letter:

5th July 2024

Dear Chritophe,

Thank you very much for your enquiry about submitting your manuscript "EdgeHOG: fine-grained ancestral gene order inference at tree-of-life scale" to Nature Ecology & Evolution. It sounds interesting, and we would be happy to consider it for publication. However, I'm sure you'll understand that we cannot make a firm decision about whether to send the paper out to review until we have carefully read the full paper (and appropriate background literature).

In order to submit your complete manuscript to Nature Ecology & Evolution, please use the link below:

Link Redacted

If you have any questions, please feel free to contact me.

[redacted]

Version 1:

Decision Letter:

24th September 2024

Dear Christophe,

Your Article, "EdgeHOG: fine-grained ancestral gene order inference at tree-of-life scale" has now been seen by three reviewers. You will see from their comments copied below that while they find your work of considerable potential interest, they have raised quite substantial concerns that must be addressed. In light of these comments, we cannot accept the manuscript for publication, but would be very interested in considering a revised version that addresses these serious concerns.

We hope you will find the reviewers' comments useful as you decide how to proceed. If you wish to submit a substantially revised manuscript, please bear in mind that we will be reluctant to approach the reviewers again in the absence of major revisions.

If you choose to revise your manuscript taking into account all reviewer and editor comments, please highlight all changes in the manuscript text file [OPTIONAL: in Microsoft Word format].

* Include a "Response to reviewers" document detailing, point-by-point, how you addressed each referee comment. If no action was taken to address a point, you must provide a compelling argument. This response will be sent back to the referees along with the revised manuscript.

* If you have not done so already we suggest that you begin to revise your manuscript so that it conforms to our Article format instructions at <http://www.nature.com/natecolevol/info/final-submission>. Refer also to any guidelines provided in this letter.

Link Redacted

If you wish to submit a suitably revised manuscript we would hope to receive it within 6 months. If you cannot send it within this time, please let us know. We will be happy to consider your revision so long as nothing similar has been accepted for publication at Nature Ecology & Evolution or published elsewhere.

Nature Ecology & Evolution is committed to improving transparency in authorship. As part of our efforts in this direction, we are now requesting that all authors identified as 'corresponding author' on published papers create and link their Open Researcher and Contributor Identifier (ORCID) with their account on the Manuscript Tracking System (MTS), prior to acceptance. This applies to primary research papers only. ORCID helps the scientific community achieve unambiguous attribution of all scholarly contributions. You can create and link your ORCID from the home page of the MTS by clicking on 'Modify my Springer Nature account'. For more information please visit www.springernature.com/orcid.

Thank you for the opportunity to review your work.

[redacted]

Reviewer expertise:

Reviewer #1: bioinformatics, evolution of vertebrate genomes

Reviewer #2: bioinformatics, chromosome evolution

Reviewer #3: evolution of eukaryotic genomes

Reviewers' comments:

Reviewer #1 (Remarks to the Author):

The manuscript by Bernard et al. presents a new method, EdgeHOG, which reconstructs gene order in ancestral genomes. EdgeHOG operates in three main phases. In the first, a bottom-up phase, gene-gene adjacencies from extant genomes are propagated to the nearest ancestral nodes, creating graphs where genes are vertices and adjacencies are edges. Next, these "child graphs" are pruned for genes specific to each graph and further propagated up the tree, generating "parent graphs" that capture adjacencies from the starting child graphs. In the second phase, a top-down process, only evolutionary conserved edges—those observed in two child graphs or between a parent and a child graph—are retained, while others are removed. Finally, in the last step, the graphs are linearised by following edges of maximum support, resulting in ancestral contigs. After showcasing EdgeHOG's performance through benchmark experiments, the method is applied to the OMA database to compute ancestral contigs in 1,133 ancestral genomes, using 2,845 extant genomes from both prokaryotes and eukaryotes. The authors then examine the Last Eukaryotic Common Ancestor (LECA) and analyse patterns of gene adjacency conservation across extant genomes.

I have four major points about the manuscript, which together suggest that it requires further refinement, particularly in the presentation and interpretation of results. While the manuscript has merit, it will likely appeal to a more specialised audience in its current form.

1. In the first part, the authors benchmark EdgeHOG against AGORA, demonstrating that it is more accurate, sensitive, and faster, allowing it to handle larger datasets. However, I find the improvements to be marginal. I was underwhelmed by the incremental progress represented by EdgeHOG. For instance, in the simulated dataset, EdgeHOG achieves 98.9% precision (vs. 96.0% for AGORA) and 96.8% accuracy (vs. 94.9% for AGORA)—not a dramatic improvement. Moreover, EdgeHOG does not account for gene transcriptional orientation, a significant challenge that AGORA addresses, nor does it reconstruct entire ancestral chromosomes as AGORA does. Although EdgeHOG is faster and scales better, the reported reconstruction of 1,133 ancestral genomes compared to AGORA's 624 does not seem groundbreaking, especially since a large portion of those genomes likely belong to smaller bacterial genomes (about 2/3 of the extant genomes in this study). The algorithm itself is novel and interesting, but without a clear breakthrough over existing methods, it seems to have limited appeal beyond a specialised readership.

2. The second part of the manuscript focuses on LECA, a node accessible due to the extant genomes analysed. It would be beneficial to include a species tree for the eukaryotes used in this study (even in supplementary figures) to contextualise LECA's

position relative to its descendants. In Figure 5, the authors report a significant enrichment of ancestral contigs with functionally related genes, interpreting this as indirect evidence of accurate reconstruction. However, this enrichment could also result from a technical bias—tandem duplications that independently arise in eukaryotic genomes could produce non-parsimonious adjacencies that appear ancestral and are naturally enriched for similar GO terms. A critical concern here is that the reconstructed LECA seems to contain far more genes than expected. LECA, likely derived from an archaea lineage, would not have contained more than a few thousand genes (with the current record for archaea at around 4,500 genes I believe), yet the LECA reconstructed in this study contains ~40,000 ancestral genes (rootHOGs), based on the OMA database (<https://omabrowser.org/oma/ancestralgenome/Eukaryota/syntenyl/>; this information is missing from the manuscript). This tenfold excess could result from unresolved gene duplications during the ancestral gene construction. Similarly, according to the OMA database, the Boreoeutheria ancestor (ancestor to most placental mammals) contains approximately 65,000 genes, or 3 times as much as its descendant genomes. Could the authors address this issue by providing evidence that the key findings are not biased by duplications? For example, is there a higher rate of duplications in genes showing enrichment? Can they offer evidence of parsimonious adjacency conservation in key cases? If OMA infers too many ancestral genes, do the results hold if a different orthology inference method is used?

3. The insights presented in Figure 6 lack sufficient depth. For instance, the discussion of histone clusters provides little clarity on what is novel or insightful about the LECA reconstruction. Histone clusters are well-known in metazoans, and the reference to a “new form in animals” (line 320) is unclear. Given the complex evolutionary history of histone clusters through independent tandem duplications, it is hard to see how the ancestral reconstruction can be accurate without more compelling evidence.

Additionally, the section on sex chromosomes is confusingly written. What does “215 MiY average difference” (line 332) mean? The methods section suggests each species (here, mammals) was analysed using a Mann-Whitney test, which does not involve averaging. Therefore, is 215 MiY the average difference in adjacencies of all species between autosomes and Y chromosomes? Or a difference between averages for each chromosome across species? This needs clarification. Moreover, has each autosome been tested against the genome to confirm whether this younger age of sex chromosome adjacencies is indeed a unique feature of sex chromosomes? It is worth noting that one arm of zebrafish chromosome 4 is enriched in recent pseudogene repeats, which could explain this pattern without invoking sex chromosomes.

4. As someone genuinely interested in the data EdgeHOG can produce and a potential user of the method, I found the lack of information on the 1,133 ancestral genomes disappointing. How many ancestral genes and contigs are reconstructed? What level of contiguity is achieved? Which genomes are more challenging to reconstruct? A more detailed discussion of the algorithm’s added value—specifically, which adjacencies it can reconstruct that others cannot—would be highly valuable. The method propagates ancestral genomes from younger to older nodes, raising the possibility that errors in younger genomes could be carried into older reconstructions. A discussion of error frequency and mitigation strategies would be helpful.

There are also several minor points that need to be addressed:

5. Figure 1: Shouldn’t S3.X1 and S2.X1 be inverted in the central gene tree?

6. Line 39: Muffato et al. reconstructed 624 ancestral genomes, not just 98 vertebrate genomes, as stated.

7. Line 41: AGORA has a time complexity of $O(n \times \log(n))$, not quadratic, as stated (see Muffato et al., supplementary information).

8. Is EdgeHOG parallelisable? It would be helpful to provide memory usage information compared to other methods. What computer specifications were used for runtime tests?

9. Line 317: “Varies” and “varying” in the same sentence?

10. Line 335: 24.3 MiY.

11. Line 319: “The co-localisation of cluster subunits in one locus”—isn’t this simply “a cluster”?

12. Which species tree is used? Does LECA have outgroups?

13. Supplementary tables and figures need a legend.

Reviewer #2 (Remarks to the Author):

The manuscript by Bernard et al. entitled “EdgeHOG: fine-grained ancestral gene order inference at tree-of-life scale” describes a new tool to identify ancestral gene order using previously defined HOGs. This is a very interesting tool, and it seems it supersedes AGORA in terms of speed and functionality.

I have some comments/suggestions to improve the tool and the readability of the manuscript.

Algorithm description:

All parts of the algorithm are well explained and relatively easy to follow except the ‘linearization phase’. Both the results and the methods sections for this part are confusing and difficult to follow, particularly the use of cumulative edge weights. Moreover, if I understood this correctly, I believe that Figure 2c has an issue - the edge score between gene I and gene J in the first graph has a value of 3, while in the others a value of 2, looking at the table of ‘Best neighbour for V’ I believe it should be a 2? I would suggest adding first another conflicting node, f.e. W, before explaining node V, and also colouring the paths represented in the tables.

Algorithm benchmarking:

The authors benchmarked their algorithm against AGORA using three different datasets. First, they simulated 100 genomes, then they used the YGOB, and finally 50 Vertebrate genomes with 10 of them “masked”. This is commendable and thorough. However, I would suggest the authors to include in the main text the results reported in the figures, particularly for the vertebrate set (expand lines 211-214). Moreover, the authors hint that masked species were selected depending on the polytomy of the tree and the quality of their proteomes, does this explain the broad range for both precision and recall depending on the species? It would be beneficial to add this into the text.

Seeing that the authors used two different vertebrate sets (50 species – 10 masked in Figure 3 and 156 species – 10 masked in

Fig Sup 1) and they reconstructed the gene order of several ancestral nodes. How consistent are the reconstructions between both sets? Do they reconstruct the same gene order using 50 or 156 species?

Applications:

The authors then used their reconstructions to study functional associations in ancestral contigs and also to date gene adjacencies. This particular feature of the algorithm makes it stand out and the results presented are interesting, although somehow expected. Sex chromosome evolution is an example of new chromosome configuration and rapid evolution (see for example <https://academic.oup.com/gbe/article/12/6/750/5823304>).

Discussion:

To this reviewer's view, the discussion can be more balanced, referring to all results presented. As it stands, it reads more as "Future steps", considering no citations are included. The results regarding the impact of the number genomes included in the gene order reconstruction could be added, as well as the findings relating to gene clusters (histones) and dating gene adjacencies (sex chromosomes).

The authors mentioned that one of the applications of EdgeHOGG could be "improving the assembly of extant genomes by integrating gene order knowledge from other species." How could this be achieved? Does this point refer to the benchmarking using "masked" genomes? Please expand.

Minor typos:

Line 330 – change autogametic to homogametic.

Reviewer #2 (Remarks on code availability):

I installed the tool and ran the test dataset. The instructions and explanations of the installation process and how to run the tool are very easy to follow.

I would suggest however, that the authors include a section on "dating gene adjacencies" in their github page. As far as I could see the script to do it is part of the supplementary data with the paper but not in the repository.

Reviewer #3 (Remarks to the Author):

This is an innovative manuscript, presenting the first synteny reconstruction and analysis of gene order conservation across the entire tree of life, which was achieved by using new software (EdgeHOG) that runs sufficiently fast to make the project feasible. The results are very interesting. They confirm some well-known examples such as the presence of histone gene clusters across all eukaryotes, and the scrambling of sex chromosomes, which is good to see because they serve as sanity checks. But there are also new results, particularly the evidence suggesting that LECA contained clusters of co-functional genes. Overall I am enthusiastic about this manuscript. It is well written, and in particular the description of the method (Figures 1 and 2 and their associated text) is succinct but absolutely clear. EdgeHOG appears to be a step-change in terms of synteny analysis it can achieve. The validation of EdgeHOG's performance using yeast and vertebrate data (Figure 3) is convincing.

However, I am concerned that gene adjacencies that originated by endosymbiosis are being scored incorrectly as present in LECA. The manuscript mentions this problem on lines 254 and 381 in the context of photosynthesis genes, but I think that the problem is more widespread and it may have affected the Gene Ontology analysis of functional enrichment in LECA gene clusters described on lines 244-255. I sorted the data in Table S1 by contig length to examine some of the large contigs (the ones drawn on the right side of Figure 5a). The 19-gene ribosomal protein cluster (contig 18) appears to be of chloroplast origin (four of the rp genes are annotated as chloroplastic, and secY is also a chloroplast genome gene). The 15-gene ATP synthesis cluster (contig 365) appears to be a chunk of mitochondrial genome (cox1, cytochrome b, several ATPase and NADH dehydrogenase subunits). Contigs 240 and 17 also seem to be chloroplast. I guess that some of the "nuclear" genome assemblies used in the analysis included cp and/or mt genomes (or maybe NUMTs and NUPTs annotated as nuclear genes), so that gene adjacencies in these organelle genomes are being reported as conserved across the LECA node. I think that the authors need to develop a method to systematically exclude the organelle-genome genes from the cluster detection and GO functional enrichment steps.

Minor comments

From browsing through Table S1, I also realized that it would be very helpful to provide access to some of the sequences themselves (not just GO terms), so that users could run their own BLAST searches etc. This could be done by hyperlinks to the OMA browser, or even just by providing the gene names from some model organisms. For example, I was unable to investigate whether some of the other rp genes in contig 18 are also chloroplastic, because I didn't have their sequences.

Figure 5a: It's too difficult to see the sizes of the contigs. Most of them are only 2 genes long, but they're drawn so close together that they look larger. Apart from putting the larger ones on the right, it's not obvious whether the contigs are arranged in any particular order.

Line 246: Give an indication of the range of sizes of the contigs in the text.

Line 258, reference should be to Figure 5c.

Table S3 is labelled "SuppTable1" on a tab.

Version 2:

Decision Letter:

20th May 2025

Dear Christophe,

Thank you for submitting your revised manuscript "EdgeHOG: fine-grained ancestral gene order inference at tree-of-life scale" (NATECOLEVOL-24071813B). It has now been seen again by the original reviewers and their comments are below. The reviewers find that the paper has improved in revision, and therefore we'll be happy in principle to publish it in Nature Ecology & Evolution, pending minor revisions to satisfy the reviewers' final requests and to comply with our editorial and formatting guidelines. Please note that we agree with the first reviewer that claims of advance need to be toned down, including removing the use of 'Tree of Life' scale.

Please email us a copy of the file in an editable format (Microsoft Word or LaTeX)-- we can not proceed with PDFs at this stage.

[redacted]

Reviewer #1 (Remarks to the Author):

I have read the authors' response to my comments and appreciate the effort made to address many of the points raised. Several issues have indeed been convincingly clarified, particularly with the addition of new tables and figures. However, my primary concern — scale and performance (Point #1)—remains insufficiently resolved. Given that these aspects form the core of the manuscript's claim to represent a substantial advance with edgeHOG, I believe they require more robust and transparent support.

First, the manuscript repeatedly refers to the reconstructions as being at "Tree of Life" scale. It appears this is based on the inclusion of genomes from all three domains of life — Bacteria, Archaea, and Eukaryota — in a single run. However, if the inclusion of just one genome per domain would qualify, the term becomes ambiguous. At what point does a dataset genuinely reflect "Tree of Life" scale? The real Tree of Life encompasses millions of bacterial and archaeal species and over 1.5 million described eukaryotes. As such, the current framing of this claim seems overstated. Simply sampling broadly across domains does not, in itself, constitute a dataset of sufficient scale or diversity to justify that description.

Second, it remains unclear how this study compares in scale and complexity to prior efforts. Table S1 indicates that the 2,851 extant genomes analyzed include:

- Bacteria: 1,965
- Archaea: 173
- Fungi: 246
- Metazoa: 281 (including 156 vertebrates)
- Viridiplantae: 85
- Others: 101

These numbers reveal that over 70% of the dataset comprises bacterial and archaeal genomes. While this is substantial numerically, the evolutionary relevance of this breadth is limited by the low level of conserved gene order across these lineages. Ancestral reconstructions in prokaryotes often become trivial beyond recent nodes, as synteny rapidly erodes with evolutionary time. In this context, combining Bacteria, Archaea, and complex eukaryotes in a single analysis raises questions about biological coherence.

The real challenge in ancestral genome reconstruction lies with the large, gene-rich, and structurally complex genomes of eukaryotes. In this study, 713 eukaryote genomes were analyzed (~11.5 million genes). By comparison, the earlier AGORA study analyzed 1,029 genomes totaling ~16 million genes (figures from the Genomicus websites), albeit in five separate runs due to Ensembl data partitioning. While edgeHOG processed these 713 genomes in a single run, it is unclear whether this justifies the manuscript's claim that the method represents a "milestone in genomics" (line 541). One could argue that the separation in AGORA was a pragmatic choice rather than a limitation of scalability.

In the same line of thought, the manuscript references the Earth BioGenome Project (EBP) as part of the motivation for developing edgeHOG (line 47), suggesting that the method is designed to meet the demands of such large-scale efforts, and this is used in the manuscript to reinforce the notion of scalability. Yet — as mentioned above — the actual dataset analyzed in this study is overwhelmingly dominated by prokaryotic genomes, while EBP is exclusively focused on eukaryotes. It is thus not clear

how relevant this link between edgeHOG and large-scale sequencing projects might be.

In summary, I remain concerned that the manuscript's presentation downplays the dominance of prokaryotic genomes in the dataset, which inflates the perception of scale without proportionally increasing biological complexity. Given this issue – including the rather incremental progress towards the kinds of genomic data that dominate current and future large-scale eukaryotic initiatives – I find that the framing implies a level of novelty that is not substantiated by the evidence provided.

Reviewer #2 (Remarks to the Author):

The authors addressed all my comments and went even further in a couple of them. I really appreciate the new supplementary figures showing how the benchmarking was done and how choosing the genomes might impact the reconstructions. The manuscript is now very well written and easy to follow.

I just have a very minor comment, more of a curiosity. How do the authors explain that when comparing the two reconstructions using either 50 or 156 species (new Fig S5) in the younger nodes (f.e. Human – macaque, dog-cat, dolphin-cow ancestors) they report a higher disagreement than in more internal nodes?

Reviewer #3 (Remarks to the Author):

I'm satisfied with the changes that the authors have made in response to my previous comments, particularly regarding chloroplast and mitochondrial genes. I do not request any further changes.

Version 3:

Decision Letter:

2nd July 2025

Dear Christophe,

We are pleased to inform you that your Article entitled "EdgeHOG: a method for fine-grained ancestral gene order inference at large scale", has now been accepted for publication in Nature Ecology & Evolution.

Over the next few weeks, your paper will be copyedited to ensure that it conforms to Nature Ecology and Evolution style. Once your paper is typeset, you will receive an email with a link to choose the appropriate publishing options for your paper and our Author Services team will be in touch regarding any additional information that may be required

Due to the importance of these deadlines, we ask you please us know now whether you will be difficult to contact over the next month. If this is the case, we ask you provide us with the contact information (email, phone and fax) of someone who will be able to check the proofs on your behalf, and who will be available to address any last-minute problems. Once your paper has been scheduled for online publication, the Nature press office will be in touch to confirm the details.

Acceptance of your manuscript is conditional on all authors' agreement with our publication policies (see www.nature.com/authors/policies/index.html). In particular your manuscript must not be published elsewhere and there must be no announcement of the work to any media outlet until the publication date (the day on which it is uploaded onto our web site).

Authors may need to take specific actions to achieve [compliance](https://www.springernature.com/gp/open-research/funding/policy-compliance-faqs) with funder and institutional open access mandates. If your research is supported by a funder that requires immediate open access (e.g. according to [Plan S principles](https://www.springernature.com/gp/open-research/plan-s-compliance)) then you should select the gold OA route, and we will direct you to the compliant route where possible. For authors selecting the subscription publication route, the journal's standard licensing terms will need to be accepted, including [self-archiving and license to publish](https://www.nature.com/nature-portfolio/editorial-policies/self-archiving-and-license-to-publish). Those licensing terms will supersede any other terms that the author or any third party may assert apply to any version of the manuscript.

If you have any questions about our publishing options, costs, Open Access requirements, or our legal forms, please contact

ASJournals@springernature.com

We welcome the submission of potential cover material (including a short caption of around 40 words) related to your manuscript; suggestions should be sent to Nature Ecology & Evolution as electronic files (the image should be 300 dpi at 210 x 297 mm in either TIFF or JPEG format). Please note that such pictures should be selected more for their aesthetic appeal than for their scientific content, and that colour images work better than black and white or grayscale images. Please do not try to design a cover with the Nature Ecology & Evolution logo etc., and please do not submit composites of images related to your work. I am sure you will understand that we cannot make any promise as to whether any of your suggestions might be selected for the cover of the journal.

You can generate the link yourself when you receive your article DOI by entering it here: <http://authors.springernature.com/share>.

[redacted]

P.S. Click on the following link if you would like to recommend Nature Ecology & Evolution to your librarian <http://www.nature.com/subscriptions/recommend.html#forms>

** Visit the Springer Nature Editorial and Publishing website at http://editorial-jobs.springernature.com?utm_source=ejP_NEcoE_email&utm_medium=ejP_NEcoE_email&utm_campaign=ejp_NEcoE for more information about our career opportunities. If you have any questions please click [here](mailto:editorial.publishing.jobs@springernature.com).

Open Access This Peer Review File is licensed under a Creative Commons Attribution 4.0 International License, which permits use, sharing, adaptation, distribution and reproduction in any medium or format, as long as you give appropriate credit to the original author(s) and the source, provide a link to the Creative Commons license, and indicate if changes were made. In cases where reviewers are anonymous, credit should be given to 'Anonymous Referee' and the source.

Reviewer #1 (Remarks to the Author):

The manuscript by Bernard et al. presents a new method, EdgeHOG, which reconstructs gene order in ancestral genomes. EdgeHOG operates in three main phases. In the first, a bottom-up phase, gene-gene adjacencies from extant genomes are propagated to the nearest ancestral nodes, creating graphs where genes are vertices and adjacencies are edges. Next, these "child graphs" are pruned for genes specific to each graph and further propagated up the tree, generating "parent graphs" that capture adjacencies from the starting child graphs. In the second phase, a top-down process, only evolutionary conserved edges—those observed in two child graphs or between a parent and a child graph—are retained, while others are removed. Finally, in the last step, the graphs are linearised by following edges of maximum support, resulting in ancestral contigs. After showcasing EdgeHOG's performance through benchmark experiments, the method is applied to the OMA database to compute ancestral contigs in 1,133 ancestral genomes, using 2,845 extant genomes from both prokaryotes and eukaryotes. The authors then examine the Last Eukaryotic Common Ancestor (LECA) and analyse patterns of gene adjacency conservation across extant genomes.

I have four major points about the manuscript, which together suggest that it requires further refinement, particularly in the presentation and interpretation of results. While **the manuscript has merit**, it will likely appeal to a more specialised audience in its current form.

Response: We thank Reviewer #1 for recognizing the merits of our algorithm and for the very constructive feedback, which led us to add a new functionality in the method (predicting the orientation of ancestral edges) and helped us improve the quality of the manuscript. When we benchmarked the ancestral orientation option of edgeHOG, we realized that AGORA's overall recall and precision improved upon making available the information of the orientation of extant gene adjacencies. We updated all the benchmarks accordingly, to make sure AGORA's optimal performances are fairly represented.

1. In the first part, the authors benchmark EdgeHOG against AGORA, demonstrating that it is more accurate, sensitive, and faster, allowing it to handle larger datasets. However, I find the improvements to be marginal. I was underwhelmed by the incremental progress represented by EdgeHOG. For instance, in the simulated dataset, EdgeHOG achieves 98.9% precision (vs. 96.0% for AGORA) and 96.8% accuracy (vs. 94.9% for AGORA)—not a dramatic improvement.

Response: This simulation was intended more as a sanity check than a severe challenge. Still, to address Reviewer 1's comment, we run a new simulation with more challenging parameters, with a substantial increase in the rates of gene duplications, losses, translocations and inversions (cf following table).

	"Easy" simulation	"Difficult" simulation
Rate of gene duplication	0.001	0.05
Ratio of translocation after duplication	0.2	0.2
Rate of gene loss	0.005	0.05
Rate of inversion	0.0005	0.01
Rate of translocation	0.0005	0.05

In the more challenging benchmark (see following **Figure S1**), edgeHOG's performance gains over AGORA become clearer:

“Difficult” Simulation of genome evolution. Each dot in each plot corresponds to one of the 99 ancestral levels of the species tree comprising 100 extant genomes. The x-axis gives the *Relative Evolutionary Divergence*²¹ of an ancestral node, which goes from 0 for the root to 1 for all the leaves. The y-axis of the top row gives the precision of each algorithmic step of edgeHOG and of Agora, namely the proportion of predicted edges at the ancestral level that are true edges in the corresponding “real” simulated ancestral genome. The y-axis of the bottom row gives the recall, namely the proportions of true edges that are predicted by each method.

Indeed, edgeHOG achieved a harmonic mean precision of 40.3% and recall of 18.8%, compared to AGORA’s 13.9% precision and 3.8% recall. These improvements are far more substantial than those observed in the easier simulation.

Moreover, EdgeHOG does not account for gene transcriptional orientation, a significant challenge that AGORA addresses.

Response: We thank Reviewer #1 for raising this important point. To fully address it, we have implemented this functionality, which can be activated using the `--orient_edges` option to edgeHOG, enabling it to predict the transcriptional context of each ancestral gene adjacency as codirectional ($\rightarrow\rightarrow$), convergent ($\rightarrow\leftarrow$), or divergent ($\leftarrow\rightarrow$). The same algorithms for propagating edge weights and parsimony-based edge trimming are applied, but now each context (codirectional, convergent, divergent) is assigned a specific weight in addition to the directionless weight. The most likely context for an ancestral edge is determined based on: 1) parsimony support and 2) the weight of each context.

We evaluated the performance of the `--orient_edges` option by benchmarking it against all correctly inferred edges from edgeHOG in the ancestral genome of YGOB. Our analysis revealed that 99.7% of these edges had orientations consistent with YGOB's annotation. Additionally, in the extant Vertebrata genomes reconstruction test, the mean precision of the orientation was 98.3% across the masked genomes.

Nor does EdgeHOG reconstruct entire ancestral chromosomes as AGORA does.

Response: We acknowledge in the Discussion that edgeHOG has so far not been optimised for macrosynteny and that our benchmarks focused on microsynteny. To better characterise the difference in reconstructed contigs between edgeHOG and AGORA, we have added supplementary figures describing the L50 and N50 measures of the two methods (**Figure S11**, also reproduced below for convenience), which indeed show that AGORA’s reconstructed ancestral contigs tend to be longer. However, note that longer is not necessarily more accurate: L50 and N50 can be trivially “gamed” by increasing recall at the expense of precision, leading to longer but less accurate ancestral contigs. We now reference this additional analysis when discussing the limitations of edgeHOG for reconstructing ancestral karyotypes.

112 ancestral genomes (Vertebrata Clade)

19 ancestral genomes (Yeast Gene Order Browser Clade)

Comparison of contiguity levels between AGORA and edgeHOG ancestral genomes. Each point on the scatterplots represents an ancestral genome. The first row shows inferences for the Vertebrata clade from OMA, while the second row represents yeast clades from YGOB. The first column compares the number of gene adjacencies predicted by both tools, the second compares the number of contigs, the third compares the L50 (i.e., the minimum number of longest contigs needed to cover half of the predicted adjacencies for an ancestor, based on the smaller of the total adjacencies predicted by AGORA and edgeHOG), and the last column shows the N50, which is the length of the Nth longest contig required to cover half of the adjacencies.

Although EdgeHOG is faster and scales better, the reported reconstruction of 1,133 ancestral genomes compared to AGORA’s 624 does not seem groundbreaking, especially since a large portion of those genomes likely belong to smaller bacterial genomes (about 2/3 of the extant genomes in this study). The algorithm itself is novel and interesting, but without a clear breakthrough over existing methods, it seems to have limited appeal beyond a specialized readership.

Response: We thank Reviewer #1 for stating that our algorithm is “novel and interesting”, and wish to provide additional clarifications regarding the groundbreaking nature of its scalability, which was not so obvious in our initial manuscript.

The difference of 1,133 ancestral genomes in our study (with edgeHOG) vs. 624 genomes in the AGORA paper may indeed not appear so big on the surface. However, while edgeHOG’s genomes were reconstructed in a single run, AGORA’s 624 genomes required five disjoint analyses:

- Genomicus Vertebrates: 199 ancestral genomes inferred from 200 extant species

- Genomicus Metazoa (Non-Vertebrates): 81 ancestral genomes inferred from 117 extant species
- Genomicus Fungi: 222 ancestral genomes inferred from 478 extant species
- Genomicus Plants: 59 ancestral genomes inferred from 99 extant species
- Genomicus Protist: 64 ancestral genomes inferred from 136 extant species

This is because both the input trees required by AGORA and the algorithm itself are subject to considerable scalability constraints (AGORA relies on costly pairwise comparisons of gene adjacencies, and the scalability of input trees depends on the tree inference method, but computing large and accurate input trees is typically a major bottleneck as well).

To illustrate this difference more clearly, we have expanded the analysis provided in **Figure 4**, which now shows more clearly the fundamentally different scaling behaviour.

Runtime of Agora and edgeHOG as a function of the number of input genomes

In combination with the recently released FastOMA software (Majidian et al, *Nature Methods* 2025), which also scales linearly, edgeHOG provides the first gene order reconstruction method which can simultaneously process thousands of genomes.

Thus, it is not just that edgeHOG is as or more accurate than AGORA, but it is also uniquely suited to keep up with today's and tomorrow's massive sequencing projects and unlocking their potential in comparative genomics. We have clarified this point in the introduction of the manuscript.

2. The second part of the manuscript focuses on LECA, a node accessible due to the extant genomes analyzed. It would be beneficial to include a species tree for the eukaryotes used in this study (even in supplementary figures) to contextualize LECA's position relative to its descendants.

Response: We thank Reviewer #1 for this suggestion. In response, we have added the following **Figure S7**, which illustrates the guide species tree used for the reconstruction (based on the Nov2022 release of OMA). This tree is essentially a pruned version of the NCBI taxonomy tree, containing only the genomes present in our database. The LECA node is represented as a polytomy with the following 9 children nodes: Opisthokonta (516 genomes), Viridiplantae (83 genomes), SAR

(60 genomes), *Discoba* (13 genomes), *Evosea* (7 genomes), *Metamonada* (4 genomes), *Rhodophyta* (4 genomes), *Guillardia theta* (2 genomes), and *Emiliana huxleyi* (1 genome). The node has Bacteria and Archaea as outgroups.

In Figure 5, the authors report a significant enrichment of ancestral contigs with functionally related genes, interpreting this as indirect evidence of accurate reconstruction. However, this enrichment could also result from a technical bias—tandem duplications that independently arise in eukaryotic genomes could produce non-parsimonious adjacencies that appear ancestral and are naturally enriched for similar GO terms.

Response: In our original submission, we should have made it clearer why the potential issue flagged by the reviewer is unlikely to affect our analysis: while duplicated genes exist as distinct entities after the duplication event, the inference algorithm is designed to merge them into a single hierarchical orthologous group (HOG, our model for a single ancestral gene) prior to the duplication (we improved **Figure 2** and its legend to make this point more clear). As such, if independent tandem duplications occur in two distinct eukaryotic lineages, and if the gene duplication events are well modeled by the HOGs, these duplicated genes / edges are merged into a single gene / edge by edgeHOG at the points of duplication and continue to propagate as one in preceding nodes of the species tree. Then, these tandem duplications would not result in multiple genes / edges in LECA, just in one.

To give a concrete illustrative example, despite numerous tandem duplications between histone gene adjacencies, we can observe that the ancestral contig of histone genes in LECA consists of only three gene adjacencies connecting four ancestral genes: Histones H2A, H2B, H3, and H4 (updated Table S2). There is overwhelming evidence in the literature supporting the notion that these four subunits were already present in LEGA, e.g. Hocher & Warnecke GBE 2024 doi:10.1093/gbe/evae029 state:

“In fact, the four core histones are universal to eukaryotes; there are no verified cases where even one of the four has been lost in any eukaryotic genome examined to date (Soo and Warnecke 2021). By implication, the nucleosome as a hetero-octameric complex already existed at the time of LECA.”

To carefully investigate the point raised by Reviewer #1, we decided to look for all potential cases where independent tandem duplications of gene adjacencies occurring in clades corresponding to different children nodes of LECA might have been wrongly modeled by the HOG framework as a single tandem duplication event occurring in LECA. As it turns out, only 12 of the 1009 LECA's contigs contain duplicated edges (contigs 693, 704, 707, 832, 876, 909, 910, 916, 917, 935 and 945 in **Table S3**), and none of them exhibited enrichment for specific GO terms. In addition, we found 87 gene adjacencies between modeled ancestral in-paralogs distributed in 70 contigs, but only 3 of these adjacencies contained paralogs whose GO terms contributed to the enriched process detected for the contig.

In **Figure S8** (shown here for convenience), we can clearly see that contigs with enrichment do not contain more modeled ancestral in-paralogs than contigs without enrichment.

A critical concern here is that the reconstructed LECA seems to contain far more genes than expected. LECA, likely derived from an archaea lineage, would not have contained more than a few thousand genes (with the current record for archaea at around 4,500 genes I believe), yet the LECA reconstructed in this study contains ~40,000 ancestral genes (rootHOGs), based on the OMA database (<https://omabrowser.org/oma/ancestralgenome/Eukaryota/syteny/>; this information is missing from the manuscript). This tenfold excess could result from unresolved gene duplications during the ancestral gene construction. Similarly, according to the OMA database, the Boreoeutheria ancestor (ancestor to most placental mammals) contains approximately 65,000 genes, or 3 times as much as its descendant genomes.

Response: We apologise for the lack of clarity in the initial manuscript. We only consider the collection of HOGs included within contigs as forming the ancestral genome, and not singleton HOGs (now mentioned in the manuscript). As such the LECA's reconstruction displayed in Figure 5 did not contain ~40,000 genes—but rather 3,162. Likewise, the inferred Boreoeutherian ancestor contains 21,775 genes, not ~65,000. We now state this clearly in the manuscript.

Regarding the number of HOGs at ancestral nodes, we are well aware that HOG inference is not perfect, particularly at deep nodes in the tree of life, and that the number of HOGs in an ancestral node of the species tree cannot be taken at face value. For instance, gene fragments resulting from bona fide fission events or sequencing errors can result in split HOGs, inflating the number of HOGs ascribed to a particular ancestor prior to filtering.

We transparently discussed this challenge in our latest OMA paper (<https://pubmed.ncbi.nlm.nih.gov/37962356/>). This led us to introduce, as a measure of quality, the HOG Completeness Score on the OMA browser, *i.e.* the number of species included in the HOG divided by the total number of species in the clade, thus it ranges from 0 to 1. HOGs with a low Completeness Score could be dubious as they imply a high number of losses, while high-quality HOGs typically have a Completeness Score > 0.2 (default filter on the OMA browser).

Currently, LECA contains 46,726 HOGs but only 3,557 of these HOGs have a completeness score > 0.2 . Likewise, the Boreoeutheria ancestral node contains 63,302 HOGs but only 19,334 HOGs have a Completeness score ≥ 0.2 .

Thanks to Reviewer #1 comment, we realized it was important to clarify the way the number of HOGs are displayed for each ancestor on the OMA browser. Now, each ancestral genome page displays: i) the absolute number of HOGs, ii) the number of well supported HOGs (Completeness Score > 0.2), iii) the number of HOGs included within contigs inferred by edgeHOG. These new indicators will provide the users with a better estimate of the size of the ancestral gene content (see screenshot below for the last common ancestor of Boreoeutherians).

Common name:	Boreoeutheria
NCBI taxonomic identifier:	1437010
Number of ancestral genes:	63302 (all HOGs) 19334 (well supported HOGs ?)
Number of ancestral genes included in ancestral contigs:	21755
Number of descendant extant species:	66

Interestingly, Reviewer #1's comment also prompted us to consider the number of ancestral neighbour as inferred by edgeHOG as an alternative indicator of HOG support, with HOGs without inferred neighbours (*i.e.* of degree 0) having less support than HOGs with at least one inferred neighbour (degree 1 or 2). We reasoned that spurious or highly fragmented HOGs are less likely to have inferred neighbours, because for adjacencies to be propagated up the tree of life, both adjacent HOG need to be properly inferred.

Indeed, we observe that HOGs of low Completeness scores tend to end up as singletons while HOGs of high Completeness scores tend to be included within contigs. The following new **Figure S10** clearly demonstrates this pattern for LECA's reconstruction.

Relationship between the degree (number of neighbors) of HOGs in LECA's contigs and their Completeness Score. The plot gives the distribution of Completeness Scores for HOGs at the LECA level that are included within reconstructed contigs (degree = 2), that are terminal genes in contigs (degree = 1), that are singletons and thus excluded from the ancestral genome (degree = 0). The vertical line in each ridge plot gives the median of the distribution for each degree level.

Thus, ancestral contigs inherently exclude a large number of low quality HOGs, which is why we obtain robust results without the need to explicitly exclude ancestral HOGs with low Completeness score. We have added these points for consideration in the discussion section of the manuscript.

Could the authors address this issue by providing evidence that the key findings are not biased by duplications? For example, is there a higher rate of duplications in genes showing enrichment?

Response: We demonstrate that our results are not biased by duplications in three ways.

First, the second sheet of **Table S4** provides the weight of each edge in LECA, alongside the number of extant eukaryotic species possessing the exact edge (representing the minimal possible weight). The weight of an edge may exceed the number of species with that edge due to: (i) the edge appearing multiple times within one or more extant species as a result of tandem duplications, or (ii) the edge being generated through the dynamic reconnection of genes flanking a central emerging gene in parent graphs. Importantly, in all cases, the actual weight of an edge in LECA is of the same order of magnitude as the minimal possible weight. This consistency underscores that the linearization process was not biased by duplication events which could have inflated the weight of some edges. Even in the case of histone adjacencies which have been duplicated multiple times in Eukaryota, the number of extant species having the edges inferred in LECA is so high that the same contig in LECA would have been inferred even when counting duplicated edges in extant species as 1.

Second, we showed above that contigs with enrichment do not contain more inferred ancestral in-paralogs than contigs without enrichment.

Third, we compared the distribution of the average copy numbers in extant species for i) HOGs showing enrichment in LECA and ii) HOGs not showing enrichment. Although the Kolmogorov-Smirnov test did find that both conditions didn't follow the same distribution, highlighting a potential bias, the differences are marginal (median = 1.22 copy numbers for HOGs showing enrichment, median = 1.13 for the others).

Average copy numbers in extant genomes for HOGs in LECA's contigs showing enrichment or not. The left plot gives the full distribution while the right plot zooms-in in the region with the highest density of points.

Can they offer evidence of parsimonious adjacency conservation in key cases?

Response: The second sheet of new **Table S4** presents the conservation of each edge inferred in LECA across extant Eukaryota within the OMA database. For example, it shows that the adjacency between histones H2A and H2B is conserved in the genomes of 446 extant eukaryotic species. We have updated the text to reference this data accordingly (**Table S4, sheet 2**).

If OMA infers too many ancestral genes, do the results hold if a different orthology inference method is used?

Response: Although OMA can infer a higher number of HOGs, our benchmarks demonstrate that the tool makes accurate choices regarding which HOGs to include within contigs (serving as proxies for ancestral genes) and which to exclude as singletons (**Figure S10**).

Interestingly, in the supplementary information of the AGORA paper, a benchmark shows that AGORA's reconstructions of the Boreoeutheria ancestor with either the ~23,000 reconciled gene trees from Compara or with the ~65,000 HOGs from OMA converted into reconciled gene trees yielded very similar ancestral gene adjacencies (using descendant human gene identifiers as anchors for comparisons). (https://static-content.springer.com/esm/art%3A10.1038%2Fs41559-022-01956-z/MediaObjects/41559_2022_1956_MOESM1_ESM.pdf (section "**Comparison between Ensembl Compara and OMA Hierarchical Orthology Groups (HOGs)**", pages 13-14)). We now mention this benchmark in the introduction of the manuscript.

Altogether, this demonstrates that the input HOGs from OMA provide reliable results. We now mention the above benchmark by the authors of AGORA in the introduction.

3. The insights presented in Figure 6 lack sufficient depth. For instance, the discussion of histone clusters provides little clarity on what is novel or insightful about the LECA reconstruction. Histone clusters are well-known in metazoans, and the reference to a "new form in animals" (line 320) is unclear. Given the complex evolutionary history of histone clusters through independent tandem duplications, it is hard to see how the ancestral reconstruction can be accurate without more compelling evidence.

Response: We believe the section as it was written was unclear. We chose to highlight clusters since it is an outstanding pattern in our dating of the age of adjacencies in metazoa: a large cluster of gene adjacencies dated back to Eukaryota. We show that this initially surprising pattern corresponds to a known phenomenon (histone clusters).

Our claim is not that we are able to reconstruct the complex evolutionary history of histone clusters, nor that the results we present are novel. Rather, we believe this result adds more weight to edgeHOG's results as it is able to correctly identify adjacencies between histone genes as a characteristic of the Eukaryotic common ancestor. Nevertheless, we also use this opportunity by surveying these adjacency in many clades, and fail to detect similarly large clusters in any other eukaryotic clade, further cementing the metazoan cluster of histones as an exception.

Regarding the topic of duplication, we would like to clarify three points.

1. **Handling of duplicated edges:** As previously mentioned, edgeHOG merges duplicated edges at the node of the species tree preceding that in which the tandem duplication is inferred to have occurred, based on the HOG framework. Thanks to this process, the contig containing histone genes is reassuringly only 4 genes-long in LECA (the different histone subunits).
2. **Propagation of adjacency age:** The age of a gene adjacency is determined and propagated in a top-down manner, moving from the root of the tree toward the leaves. As the species tree is traversed, when an adjacency between two HOGs in the top-down graph (with all edges explained by parsimony) is encountered for the first time at a node, the adjacency's last common ancestor (LCA) is assigned to that node. This information is then propagated to all descendant nodes that retain an edge between the descendant genes of the two HOGs. In the case of tandem duplications, the same LCA is assigned to both duplicated edges descending from the original adjacency. Consequently, what is highlighted in the figure depicting the karyotypes of extant eukaryotes are regions of duplicated edges whose ancestral edge can be traced back to LECA.
3. **Adjacencies in histone clusters that are not dated back to LECA.** In histone clusters, any edge between in-paralogs descending from the same common ancestral HOG in an Eukaryotic ancestor will not be dated as originating from LECA. Instead, such adjacencies will be dated to the LCA in which the first edge between the two paralogs is observed.

We now mention more clearly our aims in the manuscript, with new paragraphs in the discussion section, and we explicitly describe which adjacencies are dated back to LECA and which are predicted as more recent within histone clusters of extant Eukaryotes (with the following illustrative example now included as a panel of **Figure 6**).

Paralogous histone subunit adjacencies in Chicken's chromosome 1. Illustration of the karyotype of the *Gallus gallus* genome, including only named chromosomes, with adjacencies dated in million years based on EdgeHOG reconstruction. Squared in light blue is a zoomed in view on the histone cluster in chromosome 1. The older adjacencies (dated to the eukaryotic common ancestor) are adjacencies between different histone subunits. The cluster is intersped with more recent adjacency, which corresponds to adjacency between paralogous subunits and are indicated by arrows.

Additionally, the section on sex chromosomes is confusingly written. What does “215 MiY average difference” (line 332) mean? The methods section suggests each species (here, mammals) was analysed using a Mann-Whitney test, which does not involve averaging. Therefore, is 215 MiY the average difference in adjacencies of all species between autosomes and Y chromosomes? Or a difference between averages for each chromosome across species? This needs clarification. Moreover, has each autosome been tested against the genome to confirm whether this younger age of sex chromosome adjacencies is indeed a unique feature of sex chromosomes? It is worth noting that one arm of zebrafish chromosome 4 is enriched in recent pseudogene repeats, which could explain this pattern without invoking sex chromosomes.

Response: We apologize for the confusion. We reworked this section to clarify our points. The reviewer is correct that the Mann-Whitney does not involve averaging. Nevertheless, we chose to include the average difference in ages as a way to indicate the extent of the difference to the readers.

However, we now agree it only added confusion to the results. We removed this indication for the text as we already refer to **Table S5** where the average difference in age adjacencies, by species is included. We initially did not test the difference of autosomes to the rest of chromosomes as part of our analysis. This choice was made because we explicitly wanted to test differences between sex chromosomes and autosomes, based on our initial observations. As suggested, we now remade this analysis with additional testing for each autosome and added appropriate multi-testing correction (Bonferroni). Note that this increase in the number of tests reduces the power of our test, and thus the adjusted p-value is higher than the non-adjusted p-values we initially reported, however this does not affect our conclusions.

As the reviewer might have anticipated, some autosomes tend to have more recent adjacency than other chromosomes. It is, for example, the case of chromosome 19 in primates, the smaller microchromosomes in birds or chromosome 4 in *Drosophila* (itself a former X-chromosome turned autosome). We now discuss a few of these cases in the discussion, and argue they correspond to chromosomes with other remarkable features (GC-rich and high proportion of repetitive elements)

Given the reviewer comment, and our additional analysis, we reformulated this part of the text. While we still note the age of adjacencies in sex chromosomes as an outstanding feature, we now nuance our interpretation of the results. We also added a more complete discussion of our results, which include the point raised by the reviewer concerning Zebrafish's chromosome 4.

4. As someone genuinely interested in the data EdgeHOG can produce and a potential user of the method, I found the lack of information on the 1,133 ancestral genomes disappointing. How many ancestral genes and contigs are reconstructed? What level of contiguity is achieved? Which genomes are more challenging to reconstruct?

Response: We completely agree with the comment and appreciate the valuable suggestion. In response, we have created a new supplementary table (**Table S1**) that summarizes statistics on the number of genes, adjacencies, contigs, and contiguity levels for all extant and ancestral genomes in the OMA database.

A more detailed discussion of the algorithm's added value—specifically, which adjacencies it can reconstruct that others cannot—would be highly valuable.

Response: We again appreciate this excellent suggestion. edgeHOG notably stands out with improved linearization choices near the leaves compared to Agora and the ability to model gene emergence by dynamically reconnecting neighboring genes when a central gene is removed in a parent graph. We now mention this in the discussion.

The method propagates ancestral genomes from younger to older nodes, raising the possibility that errors in younger genomes could be carried into older reconstructions. A discussion of error frequency and mitigation strategies would be helpful.

Response: We apologize for the confusion. edgeHOG does not propagate ancestral genomes from child to parent nodes; rather, it propagates the synteny network, which consists of all edges propagated from all descendants up to the ancestral genes of the visited node. The heuristics for determining the final linearized genome are applied independently at each internal node of the species tree, without being influenced by choices made at other nodes in the tree (we now mention this in the section that briefly describes the algorithm).

There are also several minor points that need to be addressed:

5. Figure 1: Shouldn't S3.X1 and S2.X1 be inverted in the central gene tree?

Response: Reviewer #1 is correct, we have modified the labels accordingly.

6. Line 39: Muffato et al. reconstructed 624 ancestral genomes, not just 98 vertebrate genomes, as stated.

Response: Reviewer #1 is correct, we apologize for the mistake. In fact, we wanted to highlight what we thought was the largest inference performed in a single run by Agora (but we now realize it corresponds to Genomicus Vertebrates, 199 ancestral genomes from 200 extant genomes). We now mention clearly in the introduction that "Agora reconstructed 624 ancestral genomes in 5 independent runs (Genomicus Vertebrates, non-vertebrate Metazoa, Plants, Fungi and Protists)

7. Line 41: AGORA has a time complexity of $O(n \times \log(n))$, not quadratic, as stated (see Muffato et al., supplementary information).

Response: We corrected the sentence. Instead, we state that Agora relies on computationally costly pairwise comparisons of gene adjacencies between all extant genomes.

8. Is EdgeHOG parallelisable? It would be helpful to provide memory usage information compared to other methods. What computer specifications were used for runtime tests?

Response: edgeHOG is not yet parallelisable. We thank Reviewer #1 for the suggestion about memory usage. We have made **Figure S5** to show the RAM peak usage of both tools as a function of the size of the input phylogeny.

Peak RAM usage (in GB) of edgeHOG and AGORA as a function of the size of the input phylogeny

9. Line 317: "Varies" and "varying" in the same sentence?

Response: Here is the corrected sentence: The number and size of those clusters varies between species, e.g. 12 clusters of 14.75 adjacencies on average in *Bufo bufo* and one cluster of 109 adjacencies in *Drosophila melanogaster* (**Figure S9**).

10. Line 335: 24.3 MiY.

Response: This mention is no longer part of the manuscript

11. Line 319: "The co-localisation of cluster subunits in one locus"—isn't this simply "a cluster"?

Response: Here is new related sentence : Overall, our results highlight that the very old genetic colocalization of histone subunits on the same contig in LECA (**Figure 5a**) adopt a specific organization in animals where they still colocalize in the same locus, but in many copies of each subunit, probably as a result of more recent tandem duplications.

12. Which species tree is used? Does LECA have outgroups?

Response: The species tree is now shown in Figure S7; Bacteria and Archaea are the outgroups.

13. Supplementary tables and figures need a legend.

Response: We added a legend for all items of the supplementary materials.

Reviewer #2 (Remarks to the Author):

The manuscript by Bernard et al. entitled “EdgeHOG: fine-grained ancestral gene order inference at tree-of-life scale” describes a new tool to identify ancestral gene order using previously defined HOGs. **This is a very interesting tool, and it seems it supersedes AGORA in terms of speed and functionality.**

Response: We thank Reviewer #2 for the positive evaluation of our software solution for large-scale ancestral synteny inference.

I have some comments/suggestions to improve the tool and the readability of the manuscript.

Algorithm description: All parts of the algorithm are well explained and relatively easy to follow except the ‘linearization phase’. Both the results and the methods sections for this part are confusing and difficult to follow, particularly the use of cumulative edge weights. Moreover, if I understood this correctly, I believe that Figure 2c has an issue - the edge score between gene I and gene J in the first graph has a value of 3, while in the others a value of 2, looking at the table of ‘Best neighbour for V’ I believe it should be a 2? I would suggest adding first another conflicting node, f.e. W, before explaining node V, and also colouring the paths represented in the tables.

Response: We thank Reviewer #2 for all these suggestions, we made all the appropriate changes in **Figure 2** and improved the clarity of the linearization phase, both in the results and in the method sections.

Algorithm benchmarking: The authors benchmarked their algorithm against AGORA using three different datasets. First, they simulated 100 genomes, then they used the YGOB, and finally 50 Vertebrate genomes with 10 of them “masked”. **This is commendable and thorough.** However, I would suggest the authors to include in the main text the results reported in the figures, particularly for the vertebrate set (expand lines 211-214). Moreover, the authors hint that masked species were selected depending on the polytomy of the tree and the quality of their proteomes, does this explain the broad range for both precision and recall depending on the species? It would be beneficial to add this into the text.

Response: We appreciate Reviewer #2's recognition of the thoroughness of our benchmarking efforts. We now discuss with more details the results of the benchmark in the main text. Indeed, the direct ancestors of masked species are characterized by variability in both phylogenetic depth and polytomy level. One can expect that i) the deeper the direct ancestor, the more challenging its reconstruction becomes, ii) the greater the polytomy of a node, the lower the precision of the reconstruction. Furthermore, another factor influencing the precision and recall of the reconstruction of a masked species is the quality of its genome (i.e., contiguity level and gene annotation) when unmasked. Indeed, a “true” projected gene adjacency might be falsely scored as incorrect if it is absent due to high fragmentation of the genome (low contiguity score) or if the genome annotation pipeline missed too many genes (low proteome completeness score) or predicted too many “wrong” genes (low proteome consistency score). Last, the number of terminal duplications influences the benchmark as well because ancestral edges are projected only if they are between genes which have a single descendant gene in the masked species. This is why the recall of *S. trutta* is close to 0, as its genome is the result of a terminal whole genome duplication. We now address these points in the corresponding Results section. Additionally, to provide readers with deeper insights into this benchmark, we have included a new **Figure S3** that contextualizes the recall and precision achieved for each masked species in relation to the following seven indicators:

- 1) Number of terminal duplication events in the masked species
- 2) Contiguity level of the masked species (L90 score)
- 3) Completeness of the proteome of the masked species (Omark score)

- 4) Consistency of the proteome of the masked species (Omark score)
- 5) Level from root of the direct ancestor
- 6) Number of descendants of the direct ancestor
- 7) Polytomy level of direct ancestor (number of children nodes)

Extant masked genome's characteristics

Direct ancestor's characteristics

Recall and Precision for reconstructing a masked species based on the inferred gene order of its direct ancestor (first row), alongside various characteristics of the masked species (second row) and its most recent direct ancestor (third row). Terminal duplications represent the number of gene duplications inferred by OMA after the direct ancestor. *L90* indicates the minimal number of contigs required to capture 90% of the genes in the masked genome (i.e., lower *L90* values indicate higher assembly contiguity). *Genome completeness*, assessed by OMARK, estimates the proportion of expected genes compared to related species in OMA. *Genome consistency* measures the proportion of true positive genes in the proteome, using comparisons to related species as a proxy. The *level from root* of the direct ancestor refers to the number of parental nodes required to reach the ancestor from the root. *Number of descendants* refers to the number of leaves descending from the ancestral node, while the *number of child nodes* indicates the polytomy level of the node.

Seeing that the authors used two different vertebrate sets (50 species – 10 masked in Figure 3 and 156 species – 10 masked in Fig Sup 1) and they reconstructed the gene order of several ancestral nodes. How consistent are the reconstructions between both sets? Do they reconstruct the same gene order using 50 or 156 species?

Response: This is a very relevant point that we address in two parts.

First, we compared the gene adjacencies inferred from both sets for each of the 36 internal nodes in common to both trees (new **Figure S5**).

Similarity of ancestral reconstructions when using either 50 extant Vertebrata species or 156 species. Both phylogenies of 50 and 156 species have 36 ancestral nodes in common. For each ancestor in common, the pie chart gives the proportion of gene adjacencies in common to both inferences (grey), specific to the 50 species dataset (yellow) or to the 156 species dataset (blue).

Although generally consistent (average Jaccard index of 0.65, stdev = 0.14), the reconstructions from the two datasets differ in their levels of agreement across branches. Notably, inferring ancestral adjacencies from 156 Vertebrata species results in a greater number of inferred ancestral adjacencies overall, particularly for the most ancestral nodes, compared to using the dataset of 50 species.

Second, we compared the precision and recall of masked species when inferred using 156 genomes versus 50 genomes (new **Figure S4**).

Recall and Precision of EdgeHOG for reconstructing masked species based on the inferred gene order of their direct ancestor when using 50 extant species (X axes) or 156 extant species (Y axes). The species tree on the left corresponds to the phylogeny of 50 extant genomes. The 10 colored extant genomes correspond to those whose gene order is masked (and is to be inferred). The 10 colored internal levels correspond to the most direct ancestor of each masked species. The phylogeny of the 156 genomes is displayed in Figure S2.

Consistent with the observation that inference using 156 genomes yields more ancestral adjacencies, the recall achieved with 156 Vertebrate genomes is overall higher than with 50 species; however, this can slightly impact precision.

Applications: The authors then used their reconstructions to study functional associations in ancestral contigs and also to date gene adjacencies. This particular feature of the algorithm makes it stand out and the results presented are interesting, although somehow expected. Sex chromosome evolution is an example of new chromosome configuration and rapid evolution (see for example <https://academic.oup.com/gbe/article/12/6/750/5823304>).

Response: Thank you for making this comment. We agree the results are not novel, and correspond to already well documented trends. We do think it is worth including, as the patterns we analyze are clearly standing out in our data, and appear to be linked to known biological features. We believe it demonstrates the usefulness of the dating adjacency feature to be able to recover that, and that it may help users investigate newly sequenced genomes under this lens.

Thanks to reviewer #2 and other reviewer comments, we reworded the analyses to clarify the range of our results and provide more contexts to our readers - using in part the publication linked by reviewer #2.

Discussion: To this reviewer's view, the discussion can be more balanced, referring to all results presented. As it stands, it reads more as "Future steps", considering no citations are included. The results regarding the impact of the number genomes included in the gene order reconstruction could

be added, as well as the findings relating to gene clusters (histones) and dating gene adjacencies (sex chromosomes).

Response: We made the appropriate changes and the discussion now includes all points raised by Reviewer #2.

The authors mentioned that one of the applications of EdgeHOG could be “improving the assembly of extant genomes by integrating gene order knowledge from other species.” How could this be achieved? Does this point refer to the benchmarking using “masked” genomes? Please expand.

Response: We apologize for the lack of clarity. The reviewer’s interpretation is correct—we refer to the benchmark with masked genomes. In this benchmark we masked *all adjacencies* from selected extant genomes. Using the same approach, it is possible to infer adjacencies that might be missing from an extant fragmented genome, using the information provided by the reconstructed ancestral gene adjacencies. We made a Jupyter notebook to illustrate the idea. The Jupyter notebook in the figshare repository demonstrates the extension/bridging of contigs of a fragmented genome by running edgeHOG, followed by improving the assembly using the inferred linearized genome of its direct ancestor. But this should be seen as a proof of concept more than an actual method as this would probably be more relevant to incorporate not only the linearized graph of the direct ancestor, but the top-down and bottom-up graphs as well, and possibly from multiple ancestors.

Minor typos:

Line 330 – change autogametic to homogametic.

Response: Corrected.

Remarks on code availability: I installed the tool and ran the test dataset. The instructions and explanations of the installation process and how to run the tool are very easy to follow. I would suggest however, that the authors include a section on "dating gene adjacencies" in their github page. As far as I could see the script to do it is part of the supplementary data with the paper but not in the repository.

Response: We believe the reviewer is mentioning the code allowing to attribute an age in million years to adjacencies. Note that edgeHOG natively has the option to attribute timing to adjacencies, although this timing is reported as the name of the most ancient clade whose LCA is believed to have had the adjacency. The dating in actual time is only possible in rare cases where one happens to have a time-calibrated tree, for example from TimeTree. As suggested by the Reviewer #2, we now added the code to use a TimeTree to perform this dating as part of a companion script in our GitHub repository and documented it in the README.

Reviewer #3 (Remarks to the Author):

This is an innovative manuscript, presenting the first synteny reconstruction and analysis of gene order conservation across the entire tree of life, which was achieved by using new software (EdgeHOG) that runs sufficiently fast to make the project feasible. The results are very interesting. They confirm some well-known examples such as the presence of histone gene clusters across all eukaryotes, and the scrambling of sex chromosomes, which is good to see because they serve as sanity checks. But there are also new results, particularly the evidence suggesting that LECA contained clusters of co-functional genes. Overall I am enthusiastic about this manuscript. It is well written, and in particular the description of the method (Figures 1 and 2 and their associated text) is succinct but absolutely clear. EdgeHOG appears to be a step-change in terms of synteny analysis

it can achieve. The validation of EdgeHOG's performance using yeast and vertebrate data (Figure 3) is convincing.

Response: We thank the reviewer for their enthusiastic, positive overall assessment of our work.

However, I am concerned that gene adjacencies that originated by endosymbiosis are being scored incorrectly as present in LECA. The manuscript mentions this problem on lines 254 and 381 in the context of photosynthesis genes, but I think that the problem is more widespread and it may have affected the Gene Ontology analysis of functional enrichment in LECA gene clusters described on lines 244-255. I sorted the data in Table S1 by contig length to examine some of the large contigs (the ones drawn on the right side of Figure 5a). The 19-gene ribosomal protein cluster (contig 18) appears to be of chloroplast origin (four of the *rp* genes are annotated as chloroplastic, and *secY* is also a chloroplast genome gene). The 15-gene ATP synthesis cluster (contig 365) appears to be a chunk of mitochondrial genome (*cox1*, cytochrome *b*, several ATPase and NADH dehydrogenase subunits). Contigs 240 and 17 also seem to be chloroplast. I guess that some of the "nuclear" genome assemblies used in the analysis included cp and/or mt genomes (or maybe NUMTs and NUPTs annotated as nuclear genes), so that gene adjacencies in these organelle genomes are being reported as conserved across the LECA node. I think that the authors need to develop a method to systematically exclude the organelle-genome genes from the cluster detection and GO functional enrichment steps.

Response: We fully acknowledge that parsimony is not the most appropriate approach for addressing reticulate evolution, and we appreciate Reviewer #3's insightful analysis of LECA's contigs.

We confirm that Eukaryota genomes in the OMA database can include both chloroplast and mitochondrial contigs, as we aimed to provide users with comparative genomics insights into organelle genomes. Consequently, edgeHOG propagates edges from organelle contigs when applied to the OMA database. However, not all mitochondrial or chloroplast contigs in OMA genomes are explicitly annotated, making it impractical to run edgeHOG with a pre-filtering step for all organelle genomes. To address this, we performed a regular expression search on all Eukaryotic contig names in OMA to identify known Plastid/Chloroplast (CP) and Mitochondrion (MT) contigs. This allowed us to count the number of known descendant CP or MT genes—and their fraction normalized by the total number of descendant genes—for all HOGs in LECA's contig. This information is now included in the **table S3**.

Regarding LECA's reconstruction, the inference of consecutive adjacencies composed only of MT genes is rather a positive result, given the consensus that LECA possessed mitochondria. However, we acknowledge that MT genes ideally should not appear next to chromosomal genes. Thanks to our new MT fraction indicator, we confirm that this is not the trend. MT genes inferred in LECA are consecutive, not interspersed.

More problematic is the presence of edges between CP genes in LECA as chloroplasts emerged from the endosymbiosis of cyanobacteria after LECA in plants, let alone the other cases of secondary endosymbiosis. Considering the inherent limitation of the underlying orthology algorithm OMA, which does not currently model horizontal gene transfers, two potential causes may explain the erroneous propagation of these chloroplastic edges:

1. **Shared Gene Adjacencies with Cyanobacteria:** Common gene adjacencies, only between cyanobacteria and CP-containing eukaryotes.
2. **Polytomy in LECA Node:** The LECA node in the NCBI tree used for reconstruction is a polytomy with nine child nodes (Opisthokonta, Viridiplantae, SAR, Discoba, Evosea, Rhodophyta, Metamonada, Guillardia theta, and Emiliana huxleyi). Five of these can contain chloroplasts (Viridiplantae, Rhodophyta, SAR, Guillardia theta, and Emiliana huxleyi). Thus, any CP adjacencies shared between two CP-containing children may propagate to the LECA node, even if they are not shared with Cyanobacteria. This

typically fits the case of secondary endosymbiosis in some protists, which have acquired their chloroplast from other eukaryotes.

To address this, we used column 5 in the table (supporting children) to generate a new binary indicator for whether an edge was propagated exclusively from CP-containing children. Our analysis revealed that 429 of the 2151 edges reconstructed in LECA were propagated solely from CP-containing children. These may not necessarily be false positives; some might not originate from the plastid genome. As a matter of fact, we estimated that only 81 genes encompassed in these 429 edges have a known CP fraction > 1%.

Interestingly, these 429 edges provided a valuable opportunity to examine the extent to which they are shared exclusively with cyanobacteria among prokaryotes. We found that only 17 of these edges were propagated to the bottom-up graph of the parent node (Archaea-Bacteria-Eukaryota polytomy on the NCBI tree) only from the LECA and Cyanobacteria nodes. This underscores all the specific cases where parsimony in HOGs definition and edgeHOG's edge propagations have misinterpreted endosymbiosis as ancient ancestry (see updated **Table S3**). But overall, this is also reassuring as it shows that the issue of putative CP edges in LECA is more due to the NCBI tree lacking resolution than to reliance on parsimony.

Upon removing these 429 edges—which, again, is a conservative approach as not all may be false positives— while retaining the initial edge-contig mappings, we observed that 159 of the initial 194 contigs showing gene ontology (GO) term enrichment retained enrichment (updated **Table S3**).

Thus, even with this very conservative adjustment for potential mispropagation of chloroplastic edges, our primary finding—that genetic linkage is associated with functional coupling in LECA— remains robust.

We chose not to remove erroneously predicted edges in LECA, as we wanted to maintain transparency with readers regarding potential algorithmic biases but the new indicators in **Table S3** will now help readers to clearly identify contigs that contain MT and CP genes.

Minor comments

From browsing through Table S1, I also realized that it would be very helpful to provide access to the some of the sequences themselves (not just GO terms), so that users could run their own BLAST searches etc. This could be done by hyperlinks to the OMA browser, or even just by providing the gene names from some model organisms. For example, I was unable to investigate whether some of the other rp genes in contig 18 are also chloroplastic, because I didn't have their sequences.

Response: We have created an archive (.zip) with the sequences of all descending genes of each HOG present on a reconstructed contig in LECA having more than two genes. In this folder, sequences of each HOG are included within a fasta file of the name "HOG_Cxxxxx.fa".

Figure 5a: It's too difficult to see the sizes of the contigs. Most of them are only 2 genes long, but they're drawn so close together that they look larger. Apart from putting the larger ones on the right, it's not obvious whether the contigs are arranged in any particular order.

Response: We completely agree with the comment. In response, we removed the contigs containing only two genes and increased the spacing between the remaining columns of contigs. We now simply display the count of contigs composed of only two genes. The new figure is displayed below.

Line 246: Give an indication of the range of sizes of the contigs in the text.

Line 258, reference should be to Figure 5c.

Table S3 is labelled "SuppTable1" on a tab.

Response: We thank Reviewer #3 for having noted these 3 points. We made the appropriated corrections

Reviewer #1 (Remarks to the Author):

I have read the authors' response to my comments and appreciate the effort made to address many of the points raised. Several issues have indeed been convincingly clarified, particularly with the addition of new tables and figures. However, my primary concern — scale and performance (Point #1)—remains insufficiently resolved. Given that these aspects form the core of the manuscript's claim to represent a substantial advance with edgeHOG, I believe they require more robust and transparent support.

First, the manuscript repeatedly refers to the reconstructions as being at “Tree of Life” scale. It appears this is based on the inclusion of genomes from all three domains of life — Bacteria, Archaea, and Eukaryota — in a

single run. However, if the inclusion of just one genome per domain would qualify, the term becomes ambiguous. At what point does a dataset genuinely reflect “Tree of Life” scale? The real Tree of Life encompasses millions of bacterial and archaeal species and over 1.5 million described eukaryotes. As such, the current framing of this claim seems overstated. Simply sampling broadly across domains does not, in itself, constitute a dataset of sufficient scale or diversity to justify that description.

Second, it remains unclear how this study compares in scale and complexity to prior efforts. Table S1 indicates that the 2,851 extant genomes analyzed include:

- Bacteria: 1,965
- Archaea: 173
- Fungi: 246
- Metazoa: 281 (including 156 vertebrates)
- Viridiplantae: 85
- Others: 101

These numbers reveal that over 70% of the dataset comprises bacterial and archaeal genomes. While this is substantial numerically, the evolutionary relevance of this breadth is limited by the low level of conserved gene order across these lineages. Ancestral reconstructions in prokaryotes often become trivial beyond recent nodes, as synteny rapidly erodes with evolutionary time. In this context, combining Bacteria, Archaea, and complex eukaryotes in a single analysis raises questions about biological coherence. The real challenge in ancestral genome reconstruction lies with the large, gene-rich, and structurally complex genomes of eukaryotes. In this study, 713 eukaryote genomes were analyzed (~11.5 million genes). By comparison, the earlier AGORA study analyzed 1,029 genomes totaling ~16 million genes (figures from the Genomicus websites), albeit in five separate runs due to Ensembl data partitioning. While edgeHOG processed these 713 genomes in a single run, it is unclear whether this justifies the manuscript’s claim that the method represents a “milestone in genomics” (line 541). One could argue that the separation in AGORA was a pragmatic choice rather than a limitation of scalability

In the same line of thought, the manuscript references the Earth BioGenome Project (EBP) as part of the motivation for developing edgeHOG (line 47), suggesting that the method is designed to meet the demands of such large-scale efforts, and this is used in the manuscript to reinforce the notion

of scalability. Yet – as mentioned above – the actual dataset analyzed in this study is overwhelmingly dominated by prokaryotic genomes, while EBP is exclusively focused on eukaryotes. It is thus not clear how relevant this link between edgeHOG and large-scale sequencing projects might be.

In summary, I remain concerned that the manuscript's presentation downplays the dominance of prokaryotic genomes in the dataset, which inflates the perception of scale without proportionally increasing biological complexity. Given this issue – including the rather incremental progress towards the kinds of genomic data that dominate current and future large-scale eukaryotic initiatives – I find that the framing implies a level of novelty that is not substantiated by the evidence provided.

Response: We thank Reviewer #1 for their positive assessment of our revisions and the relevant feedback. In response to the comment, we have removed the mention of the "tree-of-scale" from the manuscript. Instead, we now state that *edgeHOG* is a method for ancestral gene order inference at large scale.

The analysis of the OMA database serves as an example of *edgeHOG* in action, but it is not the primary basis for our claim regarding the method's scalability. Rather, this claim is supported by the scalability analysis presented in the manuscript. In our benchmark, *edgeHOG* exhibited linear scalability, completing the processing of 791 eukaryotic genomes in just 1 hour and 20 minutes, compared to 43 hours required by AGORA for the same task.

Reviewer #2 (Remarks to the Author):

The authors addressed all my comments and went even further in a couple of them. I really appreciate the new supplementary figures showing how the benchmarking was done and how choosing the genomes might impact the reconstructions. The manuscript is now very well written and easy to follow.

I just have a very minor comment, more of a curiosity. How do the authors explain that when comparing the two reconstructions using either 50 or 156 species (new Fig S5) in the younger nodes (f.e. Human – macaque, dog-cat,

dolphin-cow ancestors) they report a higher disagreement than in more internal nodes?

Response: We thank the Reviewer #2 for their enthusiastic and positive assessment of our work. An explanation could be that when the same ancestor has only 2 descendant leaves (50 genomes phylogeny) vs many (156 genomes phylogeny), the synteny graph (prior to linearization) is probably i) less complex (i.e. having less conflicting nodes and less edges) and ii) less contrasted in terms of edge weights. These two differences (which may appear less marked in older nodes) may lead to a more distinct outcome during the linearization process. We are grateful to Reviewer #2 for highlighting this pattern and will investigate it further to inform future improvements to the tool.

Reviewer #3 (Remarks to the Author):

I'm satisfied with the changes that the authors have made in response to my previous comments, particularly regarding chloroplast and mitochondrial genes. I do not request any further changes.

Response: We thank the Reviewer #3 for the positive evaluation of our revised manuscript.